# Unveiling the nature of Pt-induced anti-deactivation of Ru for alkaline hydrogen oxidation reaction

Yanyan Fang[1,3], Cong Wei[1,3], Zenan Bian[1], Xuanwei Yin[1], Bo Liu[1], Zhaohui Liu[1], Peng Chi[1], Junxin Xiao[1], Wanjie Song[1], Shuwen Niu[1], Chongyang Tang[1], Jun Liu[2], Xiaolin Ge ®[1], Tongwen Xu ®[1] & Gongming Wang ®[1] ✉

While Ru owns superior catalytic activity toward hydrogen oxidation reaction and cost advantages, the catalyst deactivation under high anodic potential range severely limits its potential to replace the Pt benchmark catalyst. Unveiling the deactivation mechanism of Ru and correspondingly developing protection strategies remain a great challenge. Herein, we develop atomic Pt-functioned Ru nanoparticles with excellent anti-deactivation feature and meanwhile employ advanced operando characterization tools to probe the underlying roles of Pt in the anti-deactivation. Our studies reveal the introduced Pt single atoms effectively prevent Ru from oxidative passivation and consequently preserve the interfacial water network for the critical H* oxidative release during catalysis. Clearly understanding the deactivation nature of Ru and Pt-induced anti-deactivation under atomic levels could provide valuable insights for rationally designing stable Ru-based catalysts for hydrogen oxidation reaction and beyond.

Under the global consensus on the key actions to address environmental and climate issues, developing hydrogen-centered energy system attracts increasing attention. As one of the key reactions in hydrogen fuel cells, hydrogen oxidation reaction (HOR) bridges power and hydrogen sectors, which significantly affects their conversion efficiencies[1]. Although proton exchange membrane fuel cells have been well developed, the extensive use of precious Pt catalysts for both HOR and oxygen reduction reaction (ORR), unavoidably increases the capital cost of stacks[2–4]. Recently, it has been demonstrated non-noble metal-based catalysts can be used as ORR catalysts in anion exchange membrane fuel cells (AEMFCs) with comparable performance of Pt[5,6]. However, the catalysts for HOR are still restricted to precious Pt-based materials[7–11]. Various other catalysts have been screened for alkaline HOR, such as Rh-[12,13], Ru-[14–16], Pd-[17,18], Ir-[19–22] and Ni-[23–27] based materials, while the activity or durability still cannot compete with Pt. Among the aforementioned catalysts, Ru with lower cost possesses even higher exchange current density than Pt in alkaline media, making it promising to replace precious Pt for reducing catalyst cost[28,29]. However,

the key limiting factor of Ru is its easy-oxidation feature under HOR potential region, which leads to rapid catalyst deactivation with the sharply decreased current density at ~0.2 V (vs. RHE)[14,28,30,31]. From the perspective of real working condition of fuel cell, the dramatically increased anodic potential under transient conditions during start-up or rapid load change will inevitably cause catalytic passivation of Ru and limit its practical application[24,32–36]. Currently, most of the studies of Ru-based catalysts for HOR focus on improving the intrinsic activity[37–43], while only a few attentions are paid to solving the catalyst deactivation. Moreover, the passivation mechanism of Ru at high potential is still ambiguous, which in turn limits the rational design of stable Ru-based materials for alkaline HOR.

To this end, it has been found that confining metallic Ru onto metal oxides[32] such as TiO2[33,44] could stabilize Ru cluster for HOR and achieve stable current density in rotating disk electrode tests. For example, according to the work reported by Zhou et al., the lattice confinement allows electron transfers from TiO2 to Ru nanoparticles, thus greatly enhancing the anti-oxidation and the subsequent anti-

[1]Department of Chemistry, University of Science and Technology of China, Hefei 230026, China. [2]Institute of Solid State Physics, Hefei Institutes of Physical Science, Chinese Academy of Sciences, Hefei 230031, China. [3]These authors contributed equally: Yanyan Fang, Cong Wei. ✉e-mail: wanggm@ustc.edu.cn

deactivation ability[33]. However, the poor conductivity of metal oxides basically leads to high ohmic losses in membrane electrode assemblies and severely limits their applications. In addition, alloying Ru with other metals such as Pt and Ir, also shows promising performance at high potential region[45–50]. Nevertheless, the atomic ratios of Pt and Ir are considerably high (higher than 10%) and the real materials are typically a mixed-phase of Ru and other metals. Based on such systems, it is hard to determine which component in the alloys is the main active site. To better understand the anti-deactivation mechanism, a feasible solution is to develop Ru-based catalysts with precisely defined local structures and environments, which could offer an ideal platform for studying structure-property relation.

Aiming at this issue, we herein designed ultrafine Ru nanoparticles modified with atomic Pt sites on carbon black support (Pt-Ru/C), which reveals superior anti-deactivation feature in HOR process and thereby serves as an ideal model for studying the role of Pt in the anti-deactivation. Experimental studies reveal that both Ru/C and Pt-Ru/C deliver much higher intrinsic HOR performance than lab-synthesized Pt/C and commercial Pt/C-com, reflected by the larger exchange current densities. More importantly, the performance decay of Ru/C at potential higher than 0.2 V is well suppressed by introducing Pt single atoms. Advanced operando synchrotron radiation(SR)-based X-ray absorption spectroscopy (XAS), operando SR-based Fourier transform infrared (SR-FTIR) spectroscopy and ab initio molecular dynamics (AIMD) simulations indicate that the deactivation of Ru during HOR is due to the broken surface water network channel for H* oxidative desorption induced by Ru surface oxidation, while the introduction of Pt can significantly suppress the Ru oxidation and well preserve the accessibility of interfacial water network. This work first reveals the essence of Ru deactivation and Pt-induced anti-deactivation toward electrochemical alkaline HOR, which could provide valuable guidance for designing Ru-based alkaline HOR catalysts.

## Results and discussion

Pt doped Ru catalysts (Fig. 1a) were synthesized via a simple impregnation-annealing process displayed in Supplementary Fig. 1. In brief, an aqueous mixture containing $RuCl_3$, $H_2PtCl_6$ and carbon black (XC-72) was evaporated to collect the powder for post annealing in $H_2$ atmosphere to obtain Pt-Ru/C catalyst. The mass loading of Ru is ~10 wt% vs. carbon black, while the percentages of Pt in Pt-Ru/C, such as 3%Pt-Ru/C, are the atomic ratios of Pt to Ru. The crystal phase, chemical states and coordination structures of the synthesized materials are thoroughly studied. Based on the X-ray diffraction (XRD) results in Supplementary Fig. 2, Ru/C sample shows the hexagonal close-packed (hcp)-type Ru crystal structure. After incorporating Pt to Ru/C, the hcp-Ru structure is preserved, while no diffraction peak of face centered cubic (fcc)-Pt is observed until the Pt atomic ratio reaches 10%, indicating the high dispersion of Pt onto Ru nanoparticles (NPs) at low Pt loadings. To concretely characterize the grain structures of Ru/C and Pt-Ru/C with low Pt percentage, high angel annular dark field scanning transmission microscopy (HAADF-STEM) is used. Figure 1b and Supplementary Fig. 3a show the HAADF-STEM images of Ru/C, which display the uniform distribution of Ru NPs on carbon black particles with the Ru NPs grain size of ~1 nm[51]. For 3%Pt-Ru/C in Fig. 1c and Supplementary Fig. 3b, no significant change is observed for the particle size or its distribution. Meanwhile, the emergence of the bright dots on the nanoparticles of 3%Pt-Ru/C in Fig. 1c is attributed to the atomically dispersed Pt atoms with higher Z contrast, while no obvious aggregation of Pt is observed, unraveling the single atomic existence of Pt on Ru NPs.

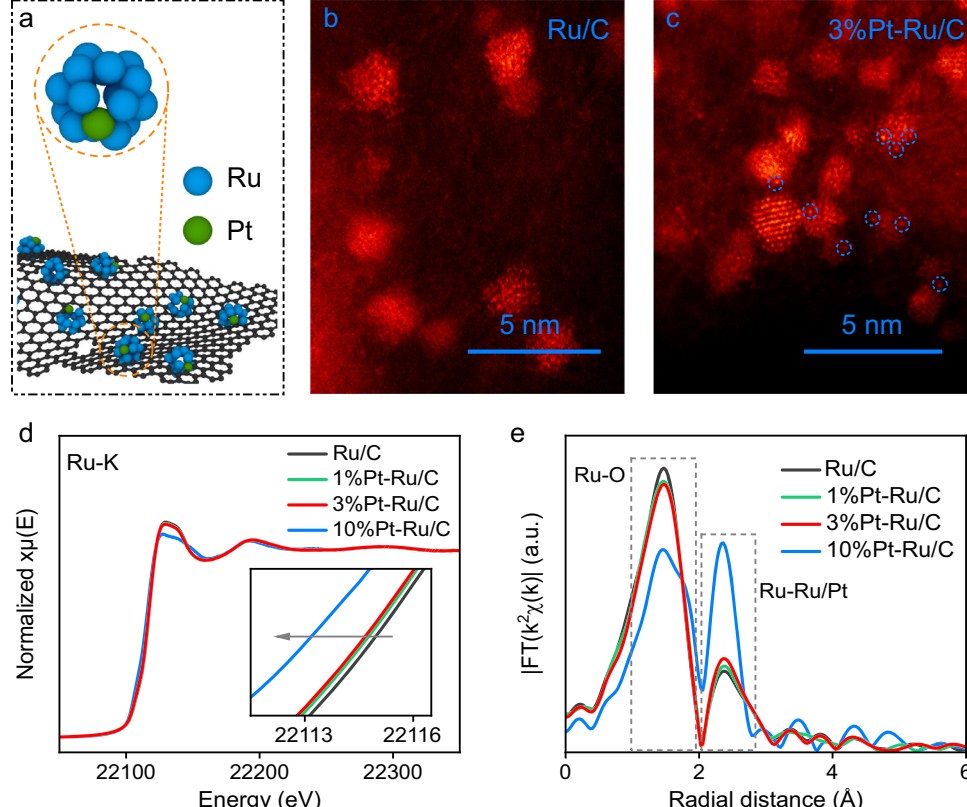

**Fig. 1 | Structural characterizations of Ru/C and Pt-Ru/C. a** Schematic diagram of Pt-Ru/C catalysts. HAADF-STEM images of Ru/C (**b**) and 3%Pt-Ru/C (**c**). Ru-K edge XANES (**d**) and EXAFS (**e**) spectra of Ru/C and 1%Pt-, 3%Pt- and 10%Pt-Ru/C.

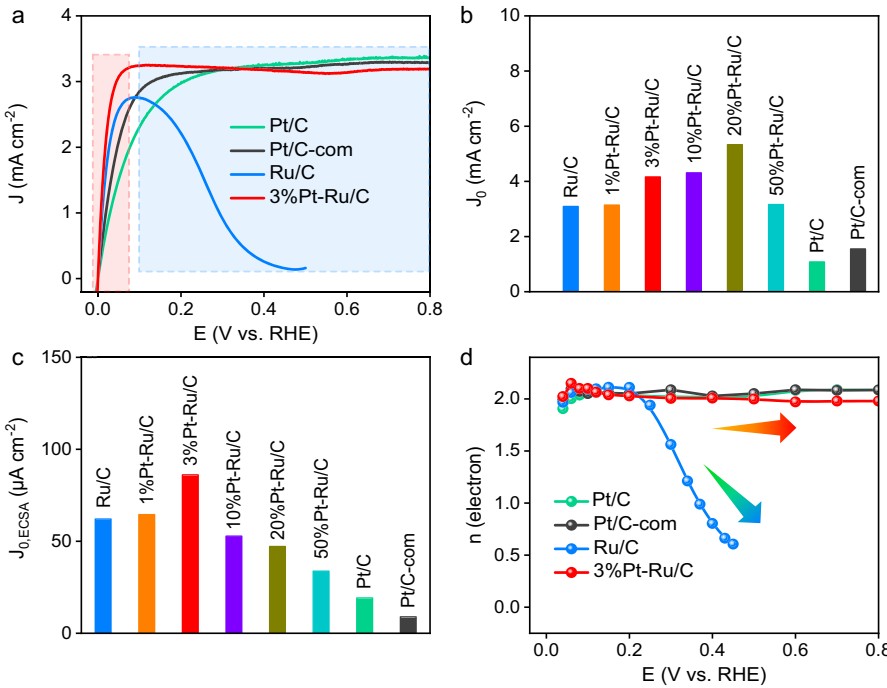

**Fig. 2 | Electrochemical hydrogen oxidation reaction in 0.1 M KOH.**
**a** Polarization curves of Ru/C, 3%Pt-Ru/C, Pt/C and Pt/C-com (Rotating speed: 2500 rpm; scan rate: 1 mV s$^{-1}$; gas flow rate: 100 mL min$^{-1}$; mass loading: 0.128 mg cm$^{-2}$; room temperature). **b** Calculated exchange current densities of different samples. **c** ECSA-normalized exchange current densities. **d** Calculated electrons involved in the HOR catalysis based on Koutecky–Levich equation.

Furthermore, the chemical states of Ru and Pt in Pt-Ru/C were investigated by XAS. Figure 1d displays the Ru-K edge X-ray absorption near edge structure (XANES) spectroscopy. The absorption edges of Ru/C and Pt-Ru/C (1%Pt, 3%Pt and 10%Pt-Ru/C) samples locate at obviously higher position than the metallic Ru foil in Supplementary Fig. 4, indicating the high valence states of Ru in Ru/C and Pt-Ru/C samples. In addition, based on the coordination structures obtained from the Ru-K edge extended X-ray absorption fine structure (EXAFS) spectra in Fig. 1e, considerable Ru-O coordination is observed in Ru/C and Pt-Ru/C samples, while the metallic feature of Ru-Ru bond becomes less significant. The oxidation of Ru originates from its exposure to air, where the ultrafine Ru particles are easily oxidized. Interestingly, after introducing Pt, the redshift of the absorption edge together with the decreased Ru-O bond and the increased Ru-Ru/Pt bond indicates the decreased valence state of Ru. Moreover, with the increased Pt loading, the oxidation degree of Ru is further decreased. Meanwhile, Pt-L$_3$ edges of the studied Pt-Ru/C samples are also analyzed, as shown in Supplementary Figs. 5 and 6. The high white line peak intensities of Pt-L$_3$ edge XANES indicate the Pt atoms in Pt-Ru/C samples are highly oxidized. However, with the increased Pt loading, the white line peak intensity is decreased, suggesting the lowered valence state of Pt. These results clearly indicate that Pt incorporation could enhance the anti-oxidation capability of Ru.

To study the role of Pt in the HOR performance of Pt-Ru/C catalysts, electrochemical tests were conducted by rotating disk electrode (RDE) in H$_2$-saturated 0.1 M KOH using 3-electrode system. Figure 2a shows the HOR polarization curves of Ru/C, 3%Pt-Ru/C, lab-synthesized Pt/C and commercial Pt/C-com. At low potential region (<0.05 V), which reflects the intrinsic activity, as anodic potential increases, the current density of Ru/C increases more sharply than those of Pt/C and even Pt/C-com, suggesting better HOR performance of Ru/C. To quantify the intrinsic activity, exchange current densities (J$_0$) are further obtained by linear fitting of the micro-polarization regions (−5 to 5 mV, Supplementary Fig. 7b, c)[52]. As shown in Fig. 2b,

the J$_0$ of Ru/C (3.09 ± 0.08 mA cm$^{-2}$) is higher than those of Pt/C (1.08 ± 0.02 mA cm$^{-2}$) and Pt/C-com (1.55 ± 0.09 mA cm$^{-2}$), indicating more favorable HOR kinetics on Ru/C. In comparison with Ru/C, 3%Pt-Ru/C shows pretty sharp current density increase at low potential range, and strikingly superior exchange current density up to 4.16 ± 0.07 mA cm$^{-2}$, demonstrating Pt modification could further enhance the intrinsic HOR activity. Similar trend is observed for the polarization curves collected at 1600 rpm (Supplementary Fig. 8). To more precisely unveil the influence of Pt doping on the per-site intrinsic activity, the electrochemical surface area (ECSA) was obtained by Cu underpotential deposition (Cu$_{upd}$)-stripping method (Supplementary Fig. 9) and the corresponding ECSA normalized exchange current densities (J$_{0,ECSA}$) were calculated (Fig. 2c). As Pt loading increases, a volcano-type trend is observed, where 3%Pt-Ru/C shows the best ECSA-normalized performance. Further increasing Pt loading would lead to suppressed HOR performance due to the formation of Pt aggerates, demonstrating single atomic Pt doping could enhance the per-site intrinsic activity.

However, as the potential further increases (>0.1 V), the current density of Ru/C almost drops to 0 at the potential above 0.4 V, indicating the unstable catalytic activity under high potential region. In contrast, the current density of 3%Pt-Ru/C can be well maintained even at high anodic potential up to 0.8 V, indicating the presence of Pt could effectively ensure the stable HOR activity. To exclude the current contribution of Pt single atoms at high potential, Pt with the same loading is introduced onto non-active metal oxides (TiO$_2$ and CeO$_2$) and the result is displayed in Supplementary Fig. 10. Apparently, the current densities of Pt-TiO$_2$ and Pt-CeO$_2$ at high potential are far lower than 3%Pt-Ru/C, which further demonstrates the stabilized current density at high potential of 3%Pt-Ru/C mainly originates from Ru sites instead of Pt itself. Furthermore, to probe the electron transfer number of the electrochemical process, the Koutecky–Levich equation is used by measuring the polarization curves at different rotating speeds (Supplementary Fig. 11) and the results are shown in Fig. 2d[53,54].

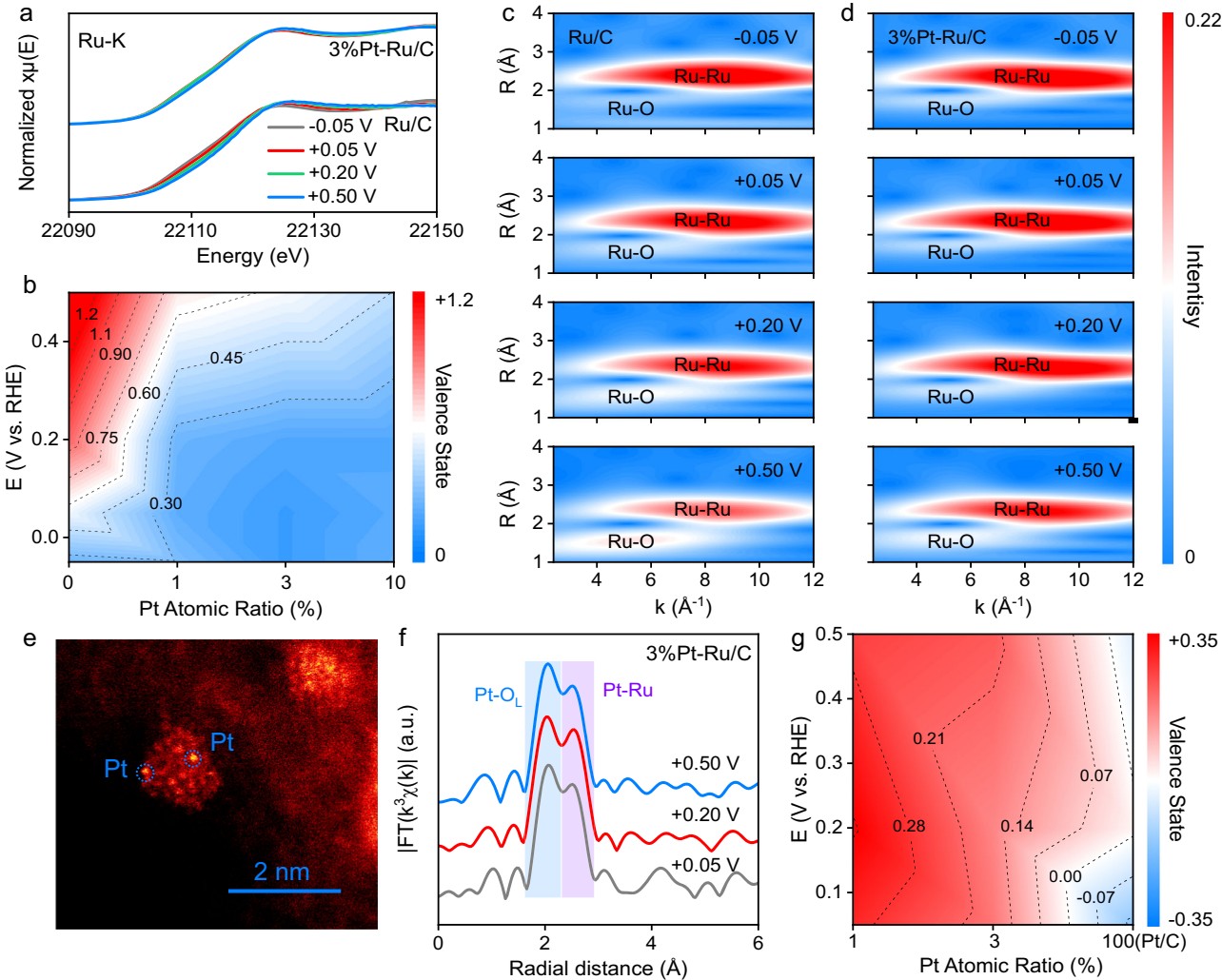

**Fig. 3 | Operando characterizations of the metal sites. a** Operando Ru-K edge XANES spectra of Ru/C and 3%Pt-Ru/C. **b** Calculated Ru valence states of Ru/C and Pt-Ru/C with different Pt loadings at various applied potentials based on the absorption edges. Ru-K edge WT-EXAFS of Ru/C (**c**) and 3%Pt-Ru/C (**d**). **e** HAADF-STEM image of 3%Pt-Ru/C pre-activated at −0.05 V. **f** Fourier-transformed (FT) Pt-L₃ edge EXAFS of 3%Pt-Ru/C. **g** Calculated Pt valence state of Pt/C and Pt-Ru/C with different Pt loadings at various applied potentials based on the white line peak intensity.

At positive potential, Pt/C and Pt/C-com undergo typical 2-electron process. Interestingly, the involved electrons in the HOR process of Ru/C decrease from ~2 at low potential (<0.1 V) to ~0 at high potential (>0.4 V), suggesting the 2-electron HOR of Ru/C is hindered at high potential region. In comparison, the 2-electron process of HOR is almost retained on 3%Pt-Ru/C, further demonstrating Pt modification can well prevent Ru from deactivation under high potential region. In addition, the Pt protection strategy is further verified by the long-term stability test. Based on the chronoamperometry test shown in Supplementary Fig. 12, after 400 min stability test, the relative current density of Ru/C shows a huge decrease of 56.5%, while the decrease of 3%Pt-Ru/C (19.3%) is much less significant, demonstrating higher stability of 3%Pt-Ru/C towards HOR catalysis. More importantly, the synthesized materials were assembled into anion exchange membrane fuel cell (AEMFC) as anode catalysts to investigate its potential in practical application (Supplementary Fig. 13). The 3%Pt-Ru/C shows not only superior AEMFC activity but also high durability with stable cell voltage output for at least 20 h, verifying the feasibility of Pt-protection strategy in real device.

To reveal the anti-deactivation nature of Pt modification, operando characterizations were operated to investigate the structural and electronic evolution of the metal sites during catalysis. The chemical state and coordination structure of Ru in Ru/C and Pt-Ru/C are monitored by operando XAS (Supplementary Fig. 14). As shown in Fig. 3a and Supplementary Fig. 15, under the potential ranging from −0.05 V to +0.50 V, the Ru-K edge spectra of Ru/C and Pt-Ru/C samples show similar contour as Ru foil (Supplementary Fig. 4a), which is essentially different from their ex-situ counterparts (Fig. 1d) or the RuO₂ and RuCl₃ standard references (Supplementary Fig. 4a), indicating the main phases of Ru/C and Pt-Ru/C are switched from the oxide-rich state to the metallic state under the studied HOR/HER region. The phase change is due to the favorable thermodynamic reduction potential of highly oxidized Ru species[55]. Moreover, it is interesting to find that the absorption edge of Ru in Ru/C obviously shifts to higher energy with increased anodic potential, while such blueshift in Pt-Ru/C system becomes less apparent. Since the absorption edge of the metal K edge is associated with the valence state, the blueshift of Ru in Ru/C suggests Ru undergoes oxidization at high HOR potential region, while the oxidation of Ru in Pt-Ru/C can be well inhibited. To more quantitatively evaluate the valence state of Ru under operando conditions, the valence state values are estimated by linearly fitting the absorption edge of the standard materials (Supplementary Fig. 4b), and the results are shown in Fig. 3b. At low potential region (<0.05 V), the valence state of Ru in Ru/C is close to metallic state, while apparent valence

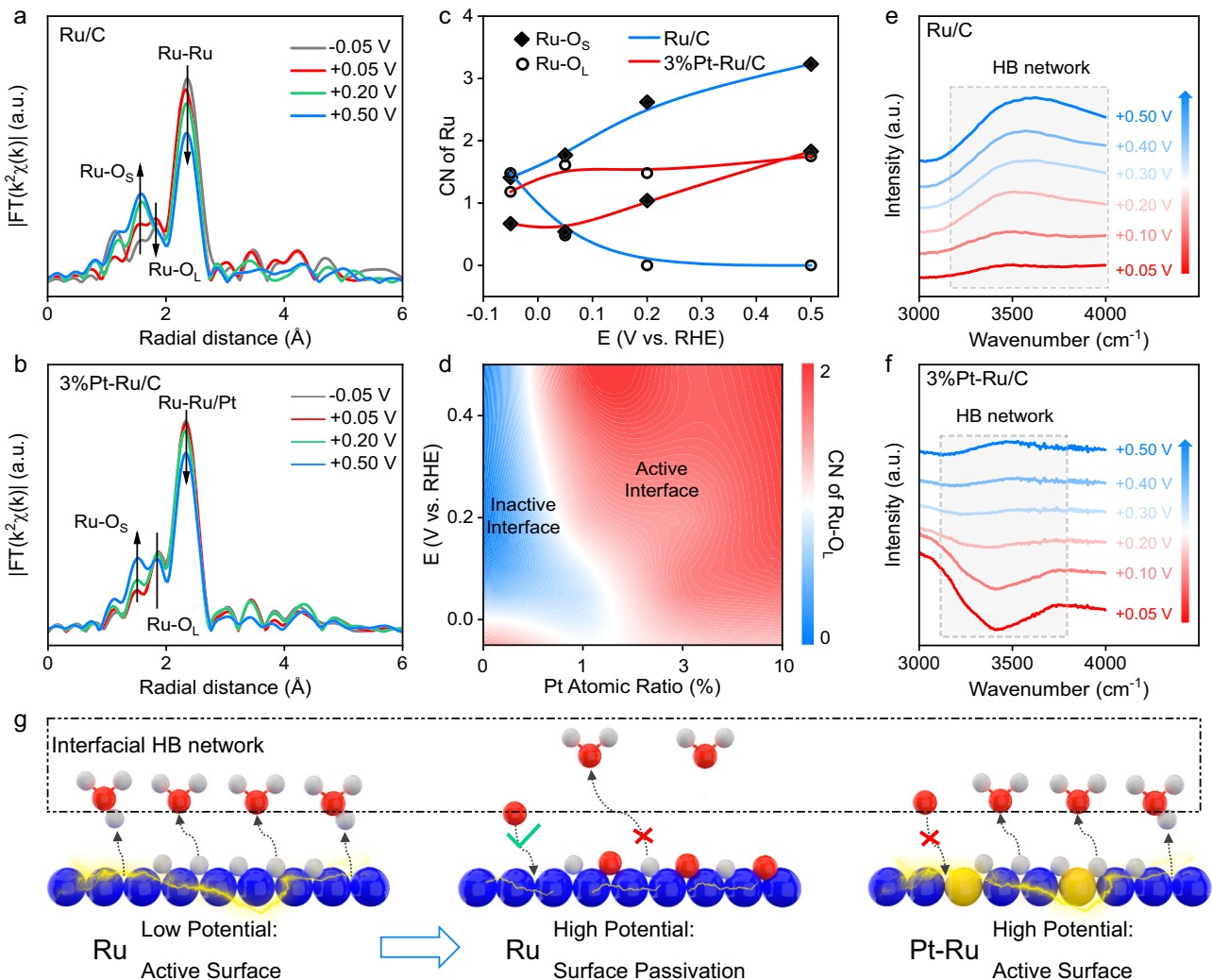

**Fig. 4 | Electrode-electrolyte interfacial structure analyses.** Operando Ru-K edge FT-EXAFS of Ru/C (**a**) and 3%Pt-Ru/C (**b**). **c** The coordination numbers (CN) of different Ru-O paths in Ru/C and 3%Pt-Ru/C obtained by fitting Ru-K edge EXAFS. **d** The CN of Ru-O$_L$ of Ru/C and Pt-Ru/C samples under various potentials. Operando SR-FTIR spectra of Ru/C (**e**) and 3%Pt-Ru/C (**f**). **g** Schematic illustration of HOR deactivation mechanism on Ru surface and the role of Pt in anti-deactivation (blue, yellow, red and light gray balls: Ru, Pt, O and H atoms).

state increase is observed under the potential higher than 0.05 V. Since the catalytic property is related to the electronic states of catalyst, it is reasonably believed that the HOR performance decay of Ru/C at high potential stems from the oxidation of Ru. Interestingly, Pt incorporation, even with 1 at% Pt loading, can suppress the oxidation of Ru. By screening the Pt loading amounts, an optimal 3%Pt-Ru/C catalyst is obtained with stabilized valence state even at potential up to 0.5 V. Interestingly, even at low potential region (<0.05 V), the oxidation state of Ru in Ru/C is still slightly higher than that in 3%Pt-Ru/C, indicating the more oxidized state of Ru/C, which is also in accordance with the smaller exchange current density of Ru/C in Fig. 2c, further demonstrating the activity of Ru is highly dependent on Ru valence state.

The essence of Ru oxidation is further investigated by analyzing the coordination structure based on the wavelet-transformed (WT) EXAFS spectra. As shown in Fig. 3c, d, compared with the standard Ru foil, RuO$_2$ and RuCl$_3$ (Supplementary Fig. 16), both Ru/C and 3%Pt-Ru/C show weak Ru-O coordination and strong Ru-Ru coordination at low potentials (<0.05 V), in accordance with the low valence state in the XANES results. However, as potential increases, the Ru-O coordination of Ru/C becomes strengthened and Ru-Ru bond is decreased, while for 3%Pt-Ru/C, the intensity increase of Ru-O bond only appears at high

potentials up to 0.5 V and Ru-Ru coordination remains almost constant. The same trend is consistently observed in 1%Pt-Ru/C and 10%Pt-Ru/C, as shown in Supplementary Fig. 17. Thus, the essence of Ru oxidation is the formation of Ru-O bond by breaking the Ru-Ru bond. It should be noted here all the studied Ru-based materials show non-negligible Ru-O coordination even at low potential, which is partly attributed to surface adsorbed O-containing active intermediates, which will be discussed later.

To reveal the role of Pt on the anti-oxidation of Ru, the chemical state of Pt is thoroughly investigated. The existence of Pt is firstly detected by HAADF-STEM and all the samples have been pre-activated at −0.05 V to exclude the influence of surface oxidation. In comparison with Ru/C (Supplementary Figs. 18 and 19), the bright dots on Ru NPs of 3%Pt-Ru/C are attributed to Pt atoms due to the higher Z contrast of Pt (Fig. 3e), indicating the single atomic existence of Pt on Ru NPs under operando operations. The Pt single atom is further verified by the operando Pt-L$_3$ edge EXAFS displayed in Fig. 3f, where the Pt-Ru bond is shorter than the Pt-Pt bond in Pt/C due to the lower atomic radius of Ru than Pt (Supplementary Fig. 20)[56]. Besides, Pt-O bond is also observed during the operando test for 3%Pt-Ru/C. The observed Pt-O bond is apparently longer than typical Pt-based oxides (such as PtO$_2$ in Supplementary Fig. 6c), indicating the elongated Pt-O bond (labeled as

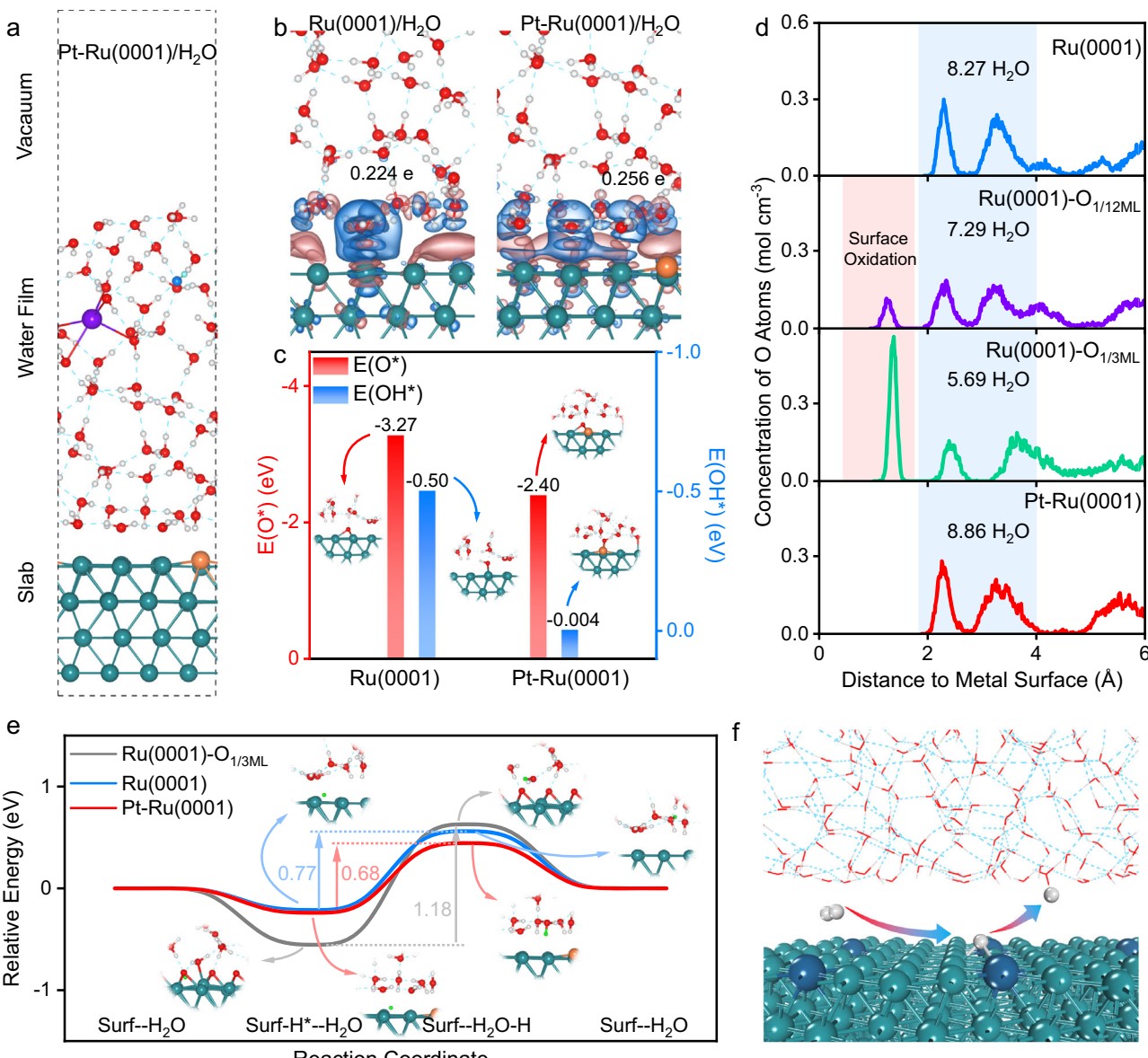

**Fig. 5 | Insights into interfacial behavior. a** Representative snapshot of the interfacial structure on Pt-Ru(0001)/$H_2O$. Gray, red, orange, turquoise balls: H, O, Pt and Ru atoms. Blue and light blue balls: O and H atoms of $OH^-$ in the electrolyte. **b** Charge density difference maps of the Ru(0001)/$H_2O$ interface and Pt-Ru(0001)/$H_2O$ interface (Isosuface value: 0.002 e $Å^{-3}$; blue: charge consumption; red: charge accumulation). **c** The adsorption energies of $O^*$ and $OH^*$ on Ru(0001) and Pt-Ru(0001) surfaces. **d** Concentration distribution profiles of O atoms along the surface normal direction. The pink shaded area represents adsorbed surface oxidation O, while the blue shaded area stands for the first two interfacial water layers within 4 Å to the corresponding surfaces. **e** Reaction pathway for a surface-adsorbed $H^*$ releasing to interfacial water layer (Green ball: the H atom involved in the reaction). **f** Schematic diagram for HOR catalysis on Pt-Ru(0001) surface.

Pt-$O_L$) is probably due to the adsorbed oxygen-containing intermediates. More accurately, the coordination structure of Pt in 3%Pt-Ru/C is quantitatively fitted with the Pt-$O_L$ and Pt-Ru paths (Supplementary Fig. 21, Supplementary Tables 1 and 3). Based on the fitting results summarized in Supplementary Fig. 22, the coordination numbers of Pt-$O_L$ (~3) and Pt-Ru (~6) in 3%Pt-Ru/C remain almost unchanged throughout the operando conditions (+0.05 V to +0.50 V), demonstrating the superior robustness of Pt sites against oxidation. Similar behavior is also observed on 1%Pt-Ru/C.

Furthermore, the electronic structures of Pt are studied by analyzing the white line peak intensity (Supplementary Fig. 23). The exact valence state is determined based on the linear fitting using the standard materials (Supplementary Fig. 6b). According to the results summarized in Fig. 3g, the valence state of Pt shows negligible dependence on the applied potential, which is due to the oxidation-

proof feature of Pt. Interestingly, as Pt loading increases, the Pt valence state is lowered, which might originate from the overall decreased oxidation degree of the metal nanoparticles, further suggesting the Pt single atom-induced reliable protection of Ru NPs against oxidation. Overall, combining the HOR performance and operando spectroscopies, the deactivation of Ru/C at high potential is attributed to severe Ru oxidation, while Pt modification could increase the anti-oxidation capability of Ru, which ensures the HOR activity at high potential. However, interesting questions arise that why and how the oxidation of Ru will induce HOR performance attenuation in the atomic level and what is the role of Pt single atom in the anti-deactivation. To answer these questions, surface adsorption structures and interfacial interactions should be analyzed.

The surface adsorption structures and interfacial interactions were probed by operando XAS and operando SR-FTIR spectroscopy.

Figure 4a, b shows the operando FT-EXAFS spectra of Ru/C and 3%Pt-Ru/C. Under operando conditions, in addition to the metallic feature of Ru-Ru coordination, there are two impressive coordination bonds, including short Ru-O bond (Ru-O$_S$) and long Ru-O bond (Ru-O$_L$). Based on the ab initio molecular dynamics (AIMD) simulations, the Ru-O$_S$ is attributed to strongly bonded O species (such as O* and OH*) due to metal site oxidation, while Ru-O$_L$ originates from weakly bonded O species (such as H$_2$O*) in the interfacial structure (Supplementary Fig. 25)[57]. For Ru/C, as anodic potential increases, due to the increased oxidation, the intensity of Ru-O$_S$ coordination is strengthened, while the surface-adsorption related Ru-O$_L$ is decreased, indicating the oxidation of Ru would inhibit the adsorption of interfacial water. In contrast, for 3%Pt-Ru/C with anti-oxidation property, the Ru-O$_L$ is preserved within the testing potential region. To more quantitatively evaluate the changes of Ru-O$_L$, the coordination numbers of different Ru-O bonds in Ru/C and 3% Pt-Ru/C are obtained by fitting the EXAFS spectra (Supplementary Figs. 26–30, Supplementary Tables 4–7). According to the fitting results in Fig. 4c, the Ru-O$_L$ of Ru/C approximately vanishes at +0.2 V, while the Ru-O$_L$ of 3%Pt-Ru/C remains almost constant. Impressively, the trend of Ru-O$_L$ could well correspond to the HOR polarization curves of Ru/C and 3%Pt-RuC, respectively, suggesting the interfacial water-related Ru-O$_L$ plays a vital role in HOR catalysis where enriched Ru-O$_L$ could ensure active interfacial structure for HOR catalysis (Fig. 4d).

To more precisely investigate the interfacial water structure, operando SR-FTIR spectroscopy with reflection mode was operated, and the acquired operando spectra at different positive potentials were background-subtracted by the spectra collected at 0 V (Supplementary Fig. 14). Based on the SR-FTIR spectra in Fig. 4e, f, the region of 3000–4000 cm$^{-1}$ is attributed to the interfacial water with hydrogen-bonding (HB) network. For Ru/C, as anodic potential increases, a positive peak at ~3500 cm$^{-1}$ emerges, while the peak intensity reaches the maximum at +0.2 V and then remains unchanged with further increasing potential, suggesting diluted interfacial water network. After introducing Pt, for 3%Pt-Ru/C, at low positive potential (<+0.3 V), the negative peaks suggest dense hydrogen network. Although at higher potential, surface water species show slightly decrease, the decrease is less significant. Considering the SR-FTIR test was conducted at the same potential region with the EXAFS test, the water adsorption behaviors of Ru/C and 3%Pt-Ru/C under operando conditions are in accordance with the trend of the Ru-O$_L$ bond in EXAFS spectra, verifying introducing Pt could prevent the oxidation-induced surface water loss at positive potential. Thus, as illustrated in Fig. 4g, the mechanism of Ru deactivation at high potential could be described as following: the severe oxidation of Ru surface is not favorable for interfacial water construction, which results in lack of interfacial water-based HB networks for H* oxidative release. The presence of Pt could prevent Ru from oxidation at high potential, and thus preserve interfacial HB network structure for efficient HOR catalysis.

To further investigate the role of interfacial water during Ru deactivation and Pt-induced Ru anti-deactivation process in alkaline HOR, AIMD and density functional theory (DFT) calculations were performed. Based on the experimental characterizations, explicit solvent structure was introduced onto Ru(0001) surface (Ru(0001)/H$_2$O) and Pt single atom-doped Ru(0001) surface (Pt-Ru(0001) /H$_2$O) (Fig. 5a and Supplementary Fig. 31). To simulate the alkaline media, OH$^-$ was further added to the solvent models and K$^+$ was introduced to balance the charge with the pH set to 14[58]. The interaction between metal surfaces and water film is investigated by analyzing the charge density differences (Fig. 5b and Supplementary Fig. 33). At the metal-water interfaces, electrons transfer from the closest two interfacial water layers to the surface. However, compared with the small charge accumulation on Ru(0001) surface (0.224 e), more electrons are localized on the Pt-Ru(0001) surface (0.256 e), which creates a more

electron-enriched surface in Pt-Ru(0001)/H$_2$O interface. The electron-enrichment on the surface might induce unfavorable affinity towards O-containing species due to the electrostatic repulsion between surface electrons and negatively charged O center (Supplementary Fig. 34). Furthermore, based on the calculated OH* and O* adsorption energies in Fig. 5c, OH* and O* are strongly adsorbed on Ru(0001)/H$_2$O interface with the adsorption energies of −0.50 and −3.27 eV, respectively (detailed structural configurations are displayed in Supplementary Fig. 35). In contrast, on Pt-Ru(0001)/H$_2$O interface, the adsorptions of OH* (−0.004 eV) and O* (−2.40 eV) are weakened, revealing the electron localization on Pt-Ru(0001) surface could weaken the adsorption of O-containing species, thus preventing surface oxidation of Ru.

To understand the influence of surface oxidation on the HOR catalytic activity of Ru, the interfacial water structures on Ru(0001) surface, Ru(0001) surfaces with different O monolayer (ML, defined per surface metal atom) coverages (Ru(0001)-O$_{1/12ML}$ and Ru(0001)-O$_{1/3ML}$) (Supplementary Fig. 31), and Pt-Ru(0001) surface are investigated. Figure 5d shows the distributions of water molecules on the studied surfaces, represented by the oxygen concentration profiles along the surface normal direction. For all the considered surfaces, the O atoms beyond 1.8 Å to surface are interfacial water molecules, while the O atoms within 1.8 Å on Ru(0001)-O$_{1/12ML}$ and Ru(0001)-O$_{1/3ML}$ surfaces are attributed to surface oxidation. Considering the electronic interaction between metal surface and water film is mainly associated with the closest two interfacial water layers, the water molecules in the first two interfacial water layers within 4 Å to the corresponding surfaces are counted. Based on the results in Fig. 5d, there are 8.27 H$_2$O molecules on Ru(0001) surface within 4 Å to the surface, while after introducing Pt onto the surface, the number of water molecules increases to 8.86, indicating denser water network on Pt-Ru(0001) surface. However, once only 1/12 ML O atom is attached on Ru(0001) surface, the water number is decreased to 7.29. Further increasing surface oxidation state to Ru(0001)-O$_{1/3ML}$ will decrease the number of water to 5.69, which is consistent with the results of the operando XAS and SR-FTIR. Thus, the decreased surface water networks induced by Ru oxidation could be the reason for the deactivation under high potential region.

Moreover, the role of interfacial water in HOR catalytic activity is probed by evaluating the critical H* desorption thermodynamics on Ru(0001), its oxidized counterpart Ru(0001)-O$_{1/3ML}$, and Pt-Ru(0001) surfaces. Figure 5e shows the reaction pathway for one H atom releasing from surface to electrolyte, which refers to the essential H* oxidative desorption in Volmer step during HOR process (detailed structural configurations are displayed in Supplementary Fig. 36). On Ru(0001) surface, the energy barrier for H* desorption is 0.77 eV, while on Pt-Ru(0001) surface, the energy is decreased to 0.68 eV, indicating more favorable H* desorption on Pt-Ru(0001) surface. The decreased energy barrier with Pt is in line with the experimental results, where Pt-Ru/C shows higher exchange current density than Ru/C. Moreover, on the oxidized Ru(0001)-O$_{1/3ML}$ surface with decreased interfacial water, the energy barrier is surged up to 1.18 eV, demonstrating the obstructed HOR process. In order to avoid the random error induced by the single water structure in calculating the H* desorption, the H* desorption thermodynamic energy barriers were calculated on the different structures during the AIMD simulations (Supplementary Figs. 37 and 42). As summarized in Supplementary Fig. 43, the total trend remains unaffected that the introduction of Pt would facilitate H* desorption while the oxidized Ru(0001)-O$_{1/3ML}$ surface always possesses hindered H* desorption. From the perspective at molecular level, the difference in H* releasing energy barrier is highly associated with the interfacial water molecule number and denser surface water content might benefit H* releasing from the surface to the electrolyte and thereby improve the HOR kinetics (Fig. 5f).

In conclusion, the mechanisms of Ru deactivation and Pt-induced anti-deactivation at high anodic potential for alkaline HOR have been thoroughly investigated by combining advanced operando spectroscopies and AIMD simulations. Under high anodic potential region, the surface of Ru NPs is easily and severely oxidized, which decreases interfacial water density and leads to unfavorable water network for H* oxidative desorption at the interface, leading to the deactivation of Ru. With atomic Pt modification, the catalyst surface becomes more oxidation-resistant, which ensures the dense interfacial water channel for H* release, resulting in stable HOR activity at high potential range. This work signifies the relations among surface oxidation, interfacial water network-induced H* desorption thermodynamics and HOR activity, which could provide valuable insights for designing efficient alkaline HOR catalysts.

## Methods

### Synthesis of Ru/C, 1%Pt-, 3%Pt-,10%Pt-, 20%Pt- and 50%Pt-Ru/C and Pt/C

100 mg XC-72 was added into 8 mL water and treated by ultrasound for 10 min, after which 0.1 mmol $RuCl_3$ was added to the aqueous mixture. The mixture was kept string for 12 h for impregnation and evaporated at 200 °C for another 16 h. The obtained black powder was annealed in $3\%H_2/Ar$ at 800 °C for 1 h. To obtain Pt doped Ru/C, controlled amount of $H_2PtCl_6$ aqueous solution was added simultaneously with $RuCl_3$. For the synthesis of Pt/C, the $RuCl_3$ solution was replaced by $H_2PtCl_6$ solution with the same concentration and volume.

### Synthesis of 0.6%Pt-, 2%Pt-TiO_2 and 0.6%Pt-, 2%Pt-CeO_2

$TiO_2$ and $CeO_2$ was synthesized using previously reported methods[59,60]. In brief, for the synthesis of $TiO_2$, the mixture containing 1.5 mL 40% hydrofluoric acid and 12.5 mL tetrabutyl titanate was transferred into 25 mL Teflon-lined stainless-steel autoclave and maintained at 180 °C for 36 h, after which the white powder was washed, dried and annealed at 300 °C for 2 h to obtain $TiO_2$. For the synthesis of $CeO_2$, the 50 mL aqueous solution containing 12.5 g NaOH and 1.085 g $Ce(NO_3)_2 \cdot 6H_2O$ was transferred into 100 mL Teflon-lined stainless-steel autoclave and maintained at 100 °C for 24 h. After washing using water, the powder was dried and annealed at 500 °C for 2 h to obtain the $CeO_2$ powder. The Pt loading on $TiO_2$ was achieved by mixing controlled amount of $H_2PtCl_6$ solution into 40 mL aqueous mixture contained 200 mg $TiO_2$ and getting stirred for 6 h. After centrifugation, the powder was dried and further annealed in $3\%H_2/Ar$ at 800 °C for 1 h. The Pt-$CeO_2$ was obtained using the same method except that $TiO_2$ was replaced by $CeO_2$.

### Materials characterizations

Powder X-ray diffraction (XRD, Philips, X'pert X-ray) patterns were obtained on a diffractometer with Cu Kα radiation (λ = 1.54182 Å). The spherical aberration corrected HAADF-STEM was conducted on JEOL JEM-ARM200F TEM/STEM with a spherical aberration corrector working at 200 kV. Hard X-ray absorption spectroscopy measurements were conducted at the beamline BL14W1 and BL11B of Shanghai Synchrotron Radiation Facility (SSRF, China) and the operando XAS test was performed in $H_2$-saturated 0.1 M KOH solution. SR-FTIR spectra were recorded at the infrared beamline BL01B of the National Synchrotron Radiation Laboratory (NSRL, China) with reflection mode and ZnSe crystal was used as the infrared transmission window (cutoff energy of ~625 cm⁻¹). The operando SR-FTIR test was carried out in $H_2$-saturated 0.1 M KOH solution and the background spectra were acquired at 0 V prior to each system measurement, and thus the negative peak accounts for the emergence or the increased amount of the chemicals while the positive one stands for the decrease.

### Electrochemical test

The electrochemical test was performed by RDE in a 5-neck H-type cell. The samples were loaded on a polished glassy carbon electrode (5 mm diameter) with the catalyst mass loading of 0.128 mg cm⁻². 0.1 M KOH electrolyte was continuously bubbled by $H_2$ gas. The polarization curves were obtained by linear sweep voltammetry (LSV) with the scan rate of 1 mV s⁻¹ at 2500 rpm rotating speed.

Exchange current densities ($J_0$) were calculated in the micro-polarization region, where the Bulter-Volmer equation can be simplified to

$$J_0 = \frac{J}{E}\frac{RT}{F} \tag{1}$$

where $J$ is the measured current density, $E$ is the potential, $R$ is the universal gas constant, $T$ is the temperature and $F$ is Faraday's constant. The exchange current density can be obtained by calculating the slope of the J-E curve in the micro-polarization regions[52].

Number of electrons involved in the reaction was determined by Koutecky–Levich equation[53,54],

$$\frac{1}{J} = \frac{1}{J_k} + \frac{1}{J_d} \tag{2}$$

where J, $J_k$ and $J_d$ are the measured current density, kinetic current density and diffusion current density, respectively. For RDE, $J_d$ could be evaluated as

$$J_d = 0.62nFD^{\frac{3}{2}}\nu^{-\frac{1}{6}}C_0\omega^{\frac{1}{2}} = BC_0\omega^{\frac{1}{2}} \tag{3}$$

where $n$ is the number of electrons involved in the reaction, $F$ is the Faraday's constant, $D$ is the diffusion coefficient of the reactant, $\nu$ is the viscosity of the electrolyte, $C_0$ is the solubility of $H_2$ in the electrolyte, $B$ is the Levich constant and $\omega$ is the rotating speed. Thus, the Koutecky–Levich equation could be rewrite as

$$\frac{1}{J} = \frac{1}{J_k} + \frac{1}{BC_0}\omega^{-\frac{1}{2}} \tag{4}$$

The number of electrons involved in the reaction could be obtained by linear fitting of the curve of $1/J$ vs. $\omega^{-1/2}$. The theoretical value of the slope ($1/BC_0$) for the two-electron HOR process is 4.87 cm² mA⁻¹ s⁻¹/². Thus, the number of the electrons involved at different potentials could be obtained as

$$n = \frac{4.87 \times 2}{\text{slope}} \tag{5}$$

The electrochemical surface area was measured by the $Cu_{upd}$-stripping voltammetry[61]. Prior to the test, the background curve was obtained in Ar-saturated 0.5 M $H_2SO_4$ electrolyte with the scan rate of 10 mV s⁻¹. The $Cu_{upd}$-stripping voltammetry was conducted in Ar-saturated 0.5 M $H_2SO_4$ containing 5 mM $CuSO_4$ with the scan rate of 10 mV s⁻¹ after the Cu underpotential deposition at 0.24 V for 3 min. The charge under the $Cu_{upd}$-stripping curve was used to determine ECSA by normalizing with a specific charge of 420 μC cm⁻².

### AEMFC tests

The synthesized materials (Ru/C and 3%Pt-Ru/C) and commercial Pt/C (75 wt% Pt) served as anode and cathode catalyst, respectively, with the both anode and cathode mass loadings of 0.3 mg cm⁻². MTCP−50 anion exchange membrane (25 um thickness) and the catalyst-coated membrane (CCM) method were used to prepare the membrane electrode assemblies (MEA) with the size of 20 mm × 20 mm[62]. For the catalyst ink preparation, 90 mg catalyst, 4.1 g water, 41 g isopropyl alcohol and 416 mg ionomer solution were mixed to form a uniform ink mixture. The catalyst ink was further sprayed onto the membrane. The CCM was immersed into 1 M KOH for 12 h before fuel cell test. For

the cell assembling, the gasket, gas diffusion layer (SGL 22BB), MEA and graphite bipolar were sandwiched. The fuel cell test was operated with the cell temperature of 85 °C, both anode and cathode humidity of 100%, symmetric back pressure of 1.5 MPa and $H_2/O_2$ flow rate of 1 L min$^{-1}$.

## Theoretical calculations

The AIMD and DFT simulations were conducted using Vienna ab initio Simulation Package (VASP) program[63,64]. The atomic core and valence electrons were expanded by projector-augmented wave (PAW) method[65] and plane-wave basis functions, and the kinetic energy cutoff was set to 400 eV. The first-order Methfessel−Paxton scheme is utilized and the Γ k-point is used in the reciprocal space of the Brillouin zone with a smearing width of 0.2 eV. The generalized gradient approximation (GGA) with the Perdew-Burke-Ernzerhof (PBE) exchange-correlation functional was employed to account for core−valence interaction[66]. The Grimme's D3 dispersion correction method with the Becke-Johnson damping function was used to consider the van der Waals dispersion forces between adsorbates and surfaces[67,68].

To construct the $Ru(0001)/H_2O$ interface, 50 water molecules are introduced on a four-layer $3 \times 4$ Ru(0001) slab. In addition, $OH^-$ was further added to the solvent models to simulate the alkaline media and $K^+$ was introduced to balance the charge with the pH set to 14 based on previously reported method[58]. The thickness of the water film is ~2.1 nm and the thickness of vacuum layer is ~1.3 nm. The size of the whole simulation box is ~4.0 nm along z axis. According to the Boltzmann distribution, the initial temperature of the samples was 100 K and the system was then heated to 300 K by a 5 ps velocity scale, followed by equilibration with a Nosé thermostat for another 15 ps at equilibrium temperature with a constant volume (Supplementary Fig. 32)[69]. The time step in all AIMD simulations was set to 1 fs, and the integration of Newton's equation was based on the Verlet algorithm implemented in VASP. For $Pt-Ru(0001)/H_2O$ system, the interfacial model is similar except that a Pt atom replaces a Ru atom on the Ru(0001) surface. For the O concentration profiles, the 10,000 structures within the 5–15 ps AIMD simulation were counted to get the average result.

In addition, to more preciously simulate the surface oxidation of Ru during HOR process, the $Ru(0001)/H_2O$ interfaces with different O atom coverages were considered (O atom coverages are 1/12 ML and 1/3 ML, respectively). Similar to the construction of $Ru(0001)/H_2O$, the configurations of the interfacial structure after O adsorption were obtained when the system reached the equilibrium during AIMD simulations.

The DFT calculation is utilized to evaluate the adsorption energies of oxygen-containing species and the oxidative desorption energies of H* in Volmer step of HOR using the structures from the equilibrated AIMD trajectories after 15 ps simulation[70]. The adsorption energy is defined as

$$E_{ads} = E_{adsorbate/surf--H_2O} - E_{adsorbate} - E_{Surf--H_2O} \qquad (6)$$

where $E_{adsorbate/Surf--H_2O}$, $E_{adsorbate}$ and $E_{Surf--H_2O}$ represent the total energies of the metal-water interface with the adsorbate, the adsorbate molecule and the bare metal-water interface, respectively. Based on the operando EXAFS and FTIR results, the Ru oxidative deactivation has huge effect on the interfacial water structure, thereby influencing the HOR performance. Therefore, the HOR mechanism on Ru surface is proposed to be that the adsorbed H* is released to the interfacial water network and will be eventually transferred to the $OH^-$ in the electric double layer through the hydrogen bond network. Thus, the initial stage of the H* discharge process when H* is released to water network is highlighted. During the calculation of H* desorption thermodynamics, a H atom is firstly adsorbed on the surface site and further

moved to the interfacial water network. The energy difference between these two processes

$$E_{des} = E_{Surf--H_2O-H} - E_{Surf-H*--H_2O} \qquad (7)$$

is determined to be the thermodynamic energy barrier for the H* desorption from the surface to the electrolyte, where $E_{Surf-H*--H_2O}$ and $E_{Surf--H_2O-H}$ represent the energies of the system with H on the surface and in the electrolyte, respectively. In order to avoid the random error induced by the single water structure during calculating the H* desorption, different structures at 9 ps and 12 ps during the AIMD simulations were also obtained, on which the H* desorption energies were calculated.

## Data availability

The data that support the findings detailed in this study are available in the Article and its Supplementary Information or from the corresponding author upon request.

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

## Acknowledgements

This work was financially supported by the National Key Research and Development Program of China (2021YFA1500400), the Natural Science Fund of China (22175163), the Natural Science Foundation of Anhui Province (2208085UD04), the Youth Innovation Promotion Association of the Chinese Academy of Science (2017483), the Fundamental Research Funds for the Central Universities (WK2060000016), the Plan for Anhui Major Provincial Science & Technology Project (Grants 2021d05050006 and 202103a05020015) and the Anhui Development and Reform Commission (AHZDCYCX-LSDT2023-08). We also thank the staff in the Shanghai Synchrotron Radiation Facility (BL14W1 and BL11B, SSRF) and the Hefei National Synchrotron Radiation Laboratory (BL01B, NSRL) for their support in XAS and SR-FTIR tests. The theoretical calculations in this paper were performed in the Supercomputing Center of University of Science and Technology of China.

## Author contributions

G.W. supervised this project. Y.F., C.W., and G.W. designed this project. Y.F., X.Y., P.C. and S.N. collected the electrochemical data. Y.F., Z.B. and J.X. conducted XAS measurements. Y.F., B.L., Z.L. and C.T. performed SR-FTIR tests. C.W. performed AIMD and DFT calculations. W.S., X.G. and T.X. provided AEMFC test facilities. J.L. provides characterization platform and resources. Y.F., C.W. and G.W. wrote the manuscript. All authors discussed and analyzed the data.

## Competing interests

The authors declare no competing interests.
