## [Peer Review File · Nature Communications]

REVIEWER COMMENTS

Reviewer #1 (Remarks to the Author):

The authors reported a mechanism study of atomic Pt doped Ru/C catalyst in resisting the well-concerned deactivation phenomenon in catalyzing the alkaline HOR at working potentials (higher than 0.2 V (vs. RHE)). Operando characterization techniques, such as SR-XAS, SR-FTIR, were employed in line with electrochemical measurement, to gain deep insight into the deactivation and anti-deactivation mechanisms at the atomic level. The most important conclusion might be that the active surface sites should be passivated by oxidatively deposited oxygen atoms which block further approaching of water molecules to the surface, while doping with Pt atoms can efficiently suppress the oxygen deposition at a wide range of working potentials, up to +0.5 V (vs. RHE). This interpretation is consistent with various experimental observations. The reviewer believed that the findings provided an important and reliable evidence for establishing a solid mechanism to rationalize the underlying mechanism, and will also be beneficial to the design of highly efficient catalysts for HOR process.

However, the reviewer suggests the authors to respond to the following concerns and make necessary revisions in the manuscript before it is acceptable for publication.

1. The authors reported an experimental exploration of alkaline HOR, in contrast, the proposed mechanism is seemed to be analyzed with an acidic HOR. For alkaline HOR, the anodic reaction should involve $H^+ + OH^- - e^- \rightarrow H_2O$, other than $H^+ + H_2O - e^- \rightarrow H_3O^+$. Accordingly, the mechanism should be redesigned up to with the real reaction process. The authors are suggested to refer to (DOI: 10.1038/NCHEM.1574 ; DOI: 10.1021/acscatal.9b00268; DOI10.1021/jacs.9b13694) for details.
2. The anti-deactivation mechanism in Pt-Ru/C catalyst has been established on the basis of suppressed oxygen deposition compared to Ru/C system. Doping with Pt was observed to successfully postpone the deposition process at least to around +0.5 V (vs.RHE). The authors placed a brief discussion in the section started at line 255, which insisted that the accumulation of electron density at the surface-layer atoms of Pt-Ru/C might be unfavorable to oxygen attachment. Beside the electron density transfer between water and metal atoms, the reviewer suggests the authors to examine also the electron transfer between Pt and Ru, to further ascertain the role of the dopant in resisting oxygen deposition.
3. The authors need to further clarify the Ru-Os and Ru-OL structures displayed in Fig4, e.g., possible structures could be, RuO*, RuOH*, as well as Ru-*OH₂, etc.. These structures can be well optimized by additional DFT calculations.
4. The interpretation (line 235-238) of Fig4e and 4f is still confusing. The HB network in the interfacial water region does be relevant to the reaction kinetics (DOI: 10.1038/nenergy.2017.31). The review suggest the author to reshape the discussion by incorporating (1) Explain why positive peak indicates “diluted interfacial water network” and negative peak indicates “dense hydrogen network”. (2) Does the slightly blue-shifting positive peak (Fig 4e) along with increased potential reveals further information of the reaction system? e.g., the HB network deformation. (3) Is there any relevance between the “diluted

interfacial water network” with the deposited oxygen atoms on metal surface? (4) Figure 5d, O distribution along surface normal direction. Slight broadening of the second peak in the case of Pt-Ru@Ru, indicating less structured HB network in this system. Does it mean that more feasible OH-migration might lead to for better HOR performance.

5. The slab models presented in Fig5 are frustrating. The geometries in the metallic region might have not been properly determined, since they are “over-ordered ”with respect to the unsymmetrical surroundings. Furthermore, it is unreasonable that Pt doping did not induce obvious structural distortion. The author should clarify this and assure other relevant descriptions should be necessarily updated accordingly, such as the adsorption energy of OHads and Oads.

6. The details of computing the reaction energy profile (Fig 5e) were not clearly described.

7. It is suggested to use “ads” or “*” uniformly, in labeling surface adsorbent.

8. “Ab initio” should be “ab initio”.

Reviewer #2 (Remarks to the Author):

In this article, Feng et al. reported Pt-induced anti-deactivation of Ru for alkaline hydrogen oxidation reaction. This is a very interesting topic to the fast-moving field of the alkaline HOR as well as anion exchange membrane fuel cells. The results presented are supported by both experimental and theory. The author successfully demonstrated anti-deactivation on Ru/C using 3% Pt by preserving Ru passivation which preserves the interfacial water network for H* oxidation. Therefore I recommend the publication of the work in the prestigious Nature Communications Journal after addressing my comments below.

1. The authors demonstrated the solid proof using XANES and EXAFS which shows reduced Ru-O which suffices the anti-deactivation provided by Pt. However, Pt is known for the excellent alkaline HOR catalyst well known in the literature which is stable under the HOR (0.0- 1.0 V vs. RHE). For instance, <https://doi.org/10.1039/C4EE00440J>. So, I am not sure the enhancement is coming from Pt itself. It looks like Pt is doing the catalysis at higher HOR potential supported by the synergistic provided by Ru. Clarify this point with an additional control experiment without HOR active components such as TiO₂ or CeO₂.

2. What about the stability of the catalyst towards long-term tests to prove the robustness of the catalysts?

3. Why the HOR polarisation is recorded at 2500 rpm and not at 1600 rpm which is typical? Explanation is needed to justify the rpm used in this study.

4. In Figure 2a, it will be good to show the full polarization curve from 0.0 V to 0.8 V to reveal the robustness of the anti-deactivation in this study.
5. In Figure 2c, the exchange current density is normalized with the geometric area and not with the electrochemical active surface area (ECSA). In order to see real improvement, the j_0 must be normalized with ECSA.
6. The reason for the improvement of the j_0 of 3% PtRu/C needs to be highlighted in greater detail.
7. Figure 2d, Nice plot. Why did the number of electron decrease with 3% PtRu/C with the increase in potential above 0.1 V vs. RHE, unlike Pt/C?
8. Have the authors explored above 10% Pt additions as the activity is increased with Pt content (Supplementary Figure S7). Please explore 20 % and 50% to see if we get the improvement further.
9. It is a good carryout deactivation strategy using Pt. Owing to the sluggish HOR kinetics in alkaline the anode loading is higher in real fuel cell devices. Therefore it is good to include such a strategy on non-noble catalysts such as TiO₂ as reported by Zhou et al. (<https://doi.org/10.1038/s41929-020-0446-9>), where similar activation is achieved using TiO₂. The author should comment on this.
10. What about the real device performance in the anion exchange membrane fuel cell data with stability to solidify the robustness strategy?
11. The operando-XANES and SR-FTIR show solid evidence of the water dynamics at the interface which gives the fundamental understanding of the de-activation of the Ru provided by Pt which appeal to the broader readership enjoyed by Nature Communications.

Reviewer #3 (Remarks to the Author):

Yanyan Fang et al. use combined experimental and computational approach to study the modifications in Ru catalyst that lead to superior catalytic activity toward hydrogen oxidation reaction (HOR) as compared to unmodified catalyst. The reason is attributed to the surface modifications via Pt single atoms that effectively prevent Ru from oxidation. It was further shown that the oxidation of a Ru catalyst disturbs interfacial water network, which is crucial for HOR reaction. While work is interesting with plethora of characterization and computational work that complements the experiments, I have several major concerns about the work.

1. While there might be evidence of superior intrinsic activity of Pt-Ru/C samples, this work provides no convincing data that proposed modification of the Ru catalyst would lead to increased durability of the electrochemical system or a device (no time studies).

2. There are several elements of the computational work that are unclear and therefore difficult to judge.

a) Why is vacuum introduced in the model (Fig 5a) and how are water molecules in a water layer prevented from drifting into the vacuum during the AIMD simulations? In a fully equilibrium state, one would expect for water molecules to distribute to the whole region above the catalyst surface.

b) How was pH taken into account?

c) When determining thermodynamics of H* desorption, was water structure obtained from equilibrated AIMD trajectories and how? Was this obtained from a snapshot and if so, from which one? Does the choice of exact water structure/water network have a huge effect on the H* desorption thermodynamics?

d) When determining barriers on Fig 5e, were and how were transition states confirmed to be transition states?

e) How are plots on Figure 5d obtained to determine the number of water molecules above the metal surfaces – from a snapshot or is this derived as an average from the production run?

f) Structures in Fig 5c and Fig 5e are very difficult to understand and it is therefore very difficult to understand the exact role of water in the HOR mechanism. Although Figure 5c and 5e look nice, it would be useful to show the same structures from these Figures but in larger format in SI.

Reviewer #4 (Remarks to the Author):

Reviewer #1 (Remarks to the Author):

The authors reported a mechanism study of atomic Pt doped Ru/C catalyst in resisting the well-concerned deactivation phenomenon in catalyzing the alkaline HOR at working potentials (higher than 0.2 V (vs. RHE)). Operando characterization techniques, such as SR-XAS, SR-FTIR, were employed in line with electrochemical measurement, to gain deep insight into the deactivation and anti-deactivation mechanisms at the atomic level. The most important conclusion might be that the active surface sites should be passivated by oxidatively deposited oxygen atoms which block further approaching of water molecules to the surface, while doping with Pt atoms can efficiently suppress the oxygen deposition at a wide range of working potentials, up to +0.5 V (vs. RHE). This interpretation is consistent with various experimental observations. The reviewer believed that the findings provided an important and reliable evidence for establishing a solid mechanism to rationalize the underlying mechanism, and will also be beneficial to the design of highly efficient catalysts for HOR process.

However, the reviewer suggests the authors to respond to the following concerns and make necessary revisions in the manuscript before it is acceptable for publication

Response: We sincerely express our gratitude to the referee for carefully reviewing the manuscript and the valuable comment, which would certainly improve the quality of the manuscript. We are also glad to see the reviewer's high evaluation of our work. According to the reviewer's comment, the point-by-point responses are presented below:

1. The authors reported an experimental exploration of alkaline HOR, in contrast, the proposed mechanism is seemed to be analyzed with an acidic HOR. For alkaline HOR, the anodic reaction should involve $H^* + OH^- - e^- \rightarrow H_2O$, other than $H^* + H_2O - e^- \rightarrow H_3O^+$. Accordingly, the mechanism should be redesigned up to with the real reaction process. The authors are suggested to refer to (DOI: 10.1038/NCHEM.1574; DOI: 10.1021/acscatal.9b00268; DOI: 10.1021/jacs.9b13694) for details.

Response: We sincerely appreciate the referee's valuable comment. It is true that the HOR should follow the alkaline mechanism " $H^* + OH^- - e^- \rightarrow H_2O$ " in alkaline media. However, until now, the effective existence of OH^- in HOR process remains unclear. Some works believe the adsorbed OH^* plays the key role in the reaction through the so-called bifunctional mechanism (Nat. Chem. 5, 300–306 (2013); Angew. Chem. Int. Ed. 56, 15594–15598 (2017); J. Am. Chem. Soc. 141, 3232–3239 (2019)), while other research works regard OH^* as reaction spectator species instead of reactive intermediate (J. Am. Chem. Soc. 2020, 142, 4985–4989; ACS Catal. 2019, 9, 6194–6201; J. Am. Chem. Soc. 2017, 139, 5156–5163).

In our work, Ru possesses strong affinity with O-containing species, and OH^- is easily adsorbed on Ru surface. Based on the AIMD simulations, the surface adsorbed OH^* possesses quite similar distance to the metal surface with the oxidation-related O^* , which indicates OH^* also forms the short Ru- O_s bond like O^* (Fig. R1). Based on our EXAFS results, the increased Ru- O_s will lead to suppressed HOR performance. Thus, OH^* might not be the reactive species in our work. In addition, based on the EXAFS and FTIR results, the Ru oxidative deactivation has huge effect on the interfacial water structure, thereby influencing the HOR performance. Therefore, we propose the HOR mechanism on Ru surface that the adsorbed H^* is released to the interfacial water network and will be eventually transferred to the OH^- in the electric double layer through the hydrogen bond network. Based on the afore-mentioned experimental results, we highlight the initial stage of the H^*

discharge process when H^* is released to water network. The calculated H^* desorption thermodynamics could correspond well with the experiment results.

Fig. R1. Concentration distribution profiles of O atoms along the surface normal direction from the AIMD simulations of Ru(0001), Ru(0001)-O_{1/12ML}, Ru(0001)-O_{1/3ML} and Ru(0001)-OH_{1/12ML}.

Moreover, to better simulate the HOR process in real condition, OH^- was further added to the solvent models to simulate the alkaline media and K^+ was introduced to balance the charge with the pH set to 14 based on previously reported method (J. Phys. Chem. Lett. 13, 10550–10557 (2022)). The results are consistent with the original one that the incorporation of Pt could protect Ru from oxidation, thereby protecting interfacial water network for the essential H^* oxidative release (Fig. R2). In the revised manuscript, we have given corresponding discussion on it.

Fig. R2. Insights into interfacial behavior. **a**, Representative snapshot of the interfacial structure on Pt-Ru(0001)/H₂O. Grey, red, orange, turquoise balls: H, O, Pt and Ru atoms. **b**, Charge density difference maps of Ru(0001)/H₂O interface and Pt-Ru(0001)/H₂O interface (Isosurface value: 0.002 e Å⁻³; blue: charge consumption; red: charge accumulation). **c**, The adsorption energies of O* and OH* on Ru(0001) and Pt-Ru(0001) surfaces. **d**, Concentration distribution profiles of O atoms along the surface normal direction. The pink shaded area represents adsorbed surface oxidation O*, while the blue shaded area stands for the first two interfacial water layers within 4 Å to the corresponding surfaces. **e**, Reaction pathway for a surface-adsorbed H* released to interfacial water layer (Green ball: the H atom involved in the reaction). **f**, Schematic diagram for HOR catalysis on Pt-Ru(0001) surface.

2. The anti-deactivation mechanism in Pt-Ru/C catalyst has been established on the basis of suppressed oxygen deposition compared to Ru/C system. Doping with Pt was observed to successfully postpone the deposition process at least to around +0.5 V (vs.RHE). The authors placed a brief discussion in the section started at line 255, which insisted that the accumulation of electron density at the surface-layer atoms of Pt-Ru/C might be unfavorable to oxygen attachment. Beside the electron density transfer between water and metal atoms, the reviewer suggests the authors to examine also the electron transfer between Pt and Ru, to further ascertain the role of the dopant in resisting oxygen deposition.

Response: We thank the referee for the valuable comment. According to the reviewer's suggestion, the electron transfer between Ru and Pt was studied to unveil the influence of O deposition. The Bader charge and the adsorption energy of OH* and O* were thoroughly considered on the surface. Based on the results illustrated in Fig. R3, compared with small charge transfer on pure Ru surface, the incorporation of Pt alters the electron density of the surrounding Ru atoms and leads to electron transfer from Ru to Pt (0.58 e), indicating enriched electron density around Pt site. Furthermore, the electron-enriched sites possess weakened adsorption of OH* and O* compared with pure Ru surface, which is in accordance with our original result that the electron enrichment is not favorable for O deposition, further demonstrating the increased electron density would benefit the surface anti-oxidation. We have added corresponding discussion on it in the revised manuscript.

Fig. R3. a, Bader charge analysis of Ru(0001) and Pt-Ru(0001) surfaces. **b**, The adsorption energies of O* and OH* on Ru(0001) and Pt-Ru(0001) surfaces.

3. The authors need to further clarify the Ru-O_s and Ru-O_L structures displayed in Fig4, e.g., possible structures could be, RuO*, RuOH*, as well as Ru-*OH₂, etc.. These structures can be well optimized by additional DFT calculations.

Response: We sincerely thank the referee for the insightful comment. To clarify the Ru-O_s and Ru-O_L, DFT calculations were performed (Fig. R4). Based on the geometric optimizations, Ru-*OH₂ possesses the longest Ru-O bond length of 2.29 Å and could be regarded as Ru-O_L, while Ru-*O possesses the shortest Ru-O bond length of 2.00 Å and is clarified as Ru-O_s. However, for Ru-*OH with the bond length of 2.16 Å, it is a little hard to determine whether it belongs to Ru-O_L or Ru-O_s. Considering that the operando test was conducted in the aqueous solution, the interfacial structure might influence the adsorption configurations. Thus, AIMD simulation results are further provided to show the difference of the mentioned three kinds of Ru-O bonds. Based on the distribution of O atoms on Ru(0001), Ru(0001)-OH_{1/12ML}, Ru(0001)-O_{1/3ML} and Ru(0001)-O_{1/12ML} illustrated in Fig. R1, the Ru-*OH₂ and Ru-*O possess the farthest and nearest O distribution, respectively. For Ru-*OH, the O distribution is quite closer to that of Ru-*O, indicating Ru-*OH could be included as Ru-O_s. In a word, the Ru-O_L mainly consists of Ru-*OH₂, while Ru-O_s contains both Ru-*O and Ru-*OH.

Fig. R4. Geometric optimizations on the adsorption configuration of H_2O^* , OH^* and O^* on Ru(0001) surface.

4. The interpretation (line 235-238) of Fig4e and 4f is still confusing. The HB network in the interfacial water region does be relevant to the reaction kinetics (DOI: 10.1038/nenergy.2017.31). The review suggest the author to reshape the discussion by incorporating (1) Explain why positive peak indicates “diluted interfacial water network” and negative peak indicates “dense hydrogen network”.

Response: We thank the referee for the comments and we are glad to clarify this issue. The operando SR-FTIR test was conducted at the infrared beamline BL01B of the National Synchrotron Radiation Laboratory (NSRL, China) with reflection mode and ZnSe crystal was used as the infrared transmission window. Prior to each test, the background spectrum was collected at 0 V vs. RHE and all the operando spectra were subtracted by the background. In this way, the negative peak accounts for the emergence or the increased amount of the chemicals, while the positive one stands for the decrease. Fig. 4e and 4f display the operando SR-FTIR results on Ru/C and 3%Pt-Ru/C. For Ru/C in Fig. 4e, as potential increases, the positive peak at $\sim 3500 \text{ cm}^{-1}$ demonstrates the decreased amount of water molecules, which is regarded as “diluted interfacial water network”. For 3%Pt-Ru/C, on the contrary, the negative peak indicates increased water content at the interface, displaying “dense hydrogen network”.

(2) Does the slightly blue-shifting positive peak (Fig 4e) along with increased potential reveals further information of the reaction system? e.g., the HB network deformation.

Response: Thank you for the valuable comment. First of all, as discussed above, the positive peak in Fig. 4e originates from the decreased interfacial water content. Then, for the blue-shift, generally, the water in higher wavenumber is attributed to the water molecules coordinated with less hydrogen bond, while the water in lower wavenumber corresponds to highly coordinated water. Thus, the blue-shifting positive peak indicates that, upon surface oxidation, more ratio of the less coordinated water is decreased. It can be explained by that the deposited surface oxygen could participate in the formation of hydrogen bond with interfacial water residue as the HB acceptor, thereby slowing down the decrease of the highly coordinated water, which results in such blue-shift. However, it does not mean the interfacial HB network is protected because both the highly coordinated water and less coordinated water are reduced and the blue-shift is due to the more severe loss of the less coordinated water. Anyway, our conclusion still matches the referee’s idea of HB network deformation due to the overall reduced interfacial water content after surface oxidation.

(3) Is there any relevance between the “diluted interfacial water network” with the deposited oxygen atoms on metal surface?

Response: The relevance between “diluted interfacial water network” with the deposited oxygen atoms on metal surface could be explained combining with the EXAFS results in Fig. 4a-4d. For Ru/C, as potential increases, the signal of deposited oxygen (Ru-O_S) is enhanced due to surface oxidation, while the interfacial water related Ru-O_L is reduced. The SR-FTIR test was conducted at the same potential region with the EXAFS test. Thus, in the SR-FTIR results, the increased anodic potential leads to more deposited surface oxygen atoms, while the positive SR-FTIR peaks demonstrate the reduced interfacial water content and the diluted interfacial water network. In a word, the diluted interfacial water network originates from the deposited oxygen atoms on the surface. We have included the discussion in the revised manuscript.

(4) Figure 5d, O distribution along surface normal direction. Slight broadening of the second peak in the case of Pt-Ru@Ru, indicating less structured HB network in this system. Does it mean that more feasible OH⁻ migration might lead to for better HOR performance.

Response: We are grateful for the referee’s thoughtful comment. Recently, the influence of OH⁻ migration on HER/HOR catalysis was investigated. For example, Qiang Sun et al. found that the addition of N-methylimidazoles could facilitate diffusion of hydroxides across the interface by holding the second layer of water close to platinum surfaces, thereby promoting the HER/HOR (Nat. Energy **8**, 859–869 (2023)). However, the influence of HB network on the OH⁻ migration is still under debate. On the one hand, Koper and coworkers found that in alkaline media, the strong interfacial electric field would lead to more rigid water structure and unfavorable OH⁻ transfer (Nat. Energy **2**, 17031 (2017)). On the other hand, Tian and coworkers unveiled that the weakened HB network would change the tetrahedral OH⁻(H₂O)₃ to the hypercoordinated OH⁻(H₂O)₄. The OH⁻ transfer through HB network mainly relies on the tetrahedral OH⁻(H₂O)₃, while the hypercoordinated OH⁻(H₂O)₄ halts the transportation (Phys. Rev. Lett. **125**, 156803(2020)).

In our work, as discussed in the response to the 1st comment, we mainly consider the H* is released to interfacial water and later reacts with OH⁻ in the electric double layer. Therefore, in this pathway, the OH⁻ migration in the second water layer might be less significant. Correspondingly, the performance enhancement with Pt doping is mainly attributed to the increased interfacial water content, which is also in accordance with our EXAFS and SR-FTIR results.

5. The slab models presented in Fig5 are frustrating. The geometries in the metallic region might have not been properly determined, since they are “over-ordered” with respect to the unsymmetrical surroundings. Furthermore, it is unreasonable that Pt doping did not induce obvious structural distortion. The author should clarify this and assure other relevant descriptions should be necessarily updated accordingly, such as the adsorption energy of OHads and Oads.

Response: We appreciate the reviewer’s carefully reviewing. In fact, although the atomic numbers of Ru (44) and Pt (78) are significantly different, the atom radius of Pt (1.39 Å) is only slightly larger than that of Ru (1.34 Å). Such small radius difference would not induce huge lattice change. We also obtained the Pt-Ru and Ru-Ru bond lengths during the AIMD simulation of Pt-Ru(0001)/H₂O system (Fig. R5). The distance between Pt and Ru atoms (2.68 Å) is only slightly longer than that between Ru and Ru (2.65 Å), which is also in accordance with our EXAFS fitting

results (Pt-Ru: ~ 2.67 Å; Ru-Ru: ~ 2.64 Å). Thus, based on our AIMD simulations and EXAFS results, the insignificant lattice change after Pt doping is acceptable.

Fig. R5. AIMD simulation of RDFs between Pt-Ru and Ru-Ru.

6. The details of computing the reaction energy profile (Fig 5e) were not clearly described.

Response: We are grateful for the reviewer's comment. Fig. 5e displays the thermodynamic process where an adsorbed H* on the metal surface is desorbed and released to the interfacial water network. During the calculation of H* desorption thermodynamics, a H atom is firstly adsorbed on the surface site and further moved to the interfacial water network. The energy difference between these two processes $E_{des} = E_{Surf--H_2O-H} - E_{Surf-H*--H_2O}$ is determined to be the thermodynamic energy barrier for the H* desorption from the surface to the electrolyte, where $E_{Surf-H*--H_2O}$ and $E_{Surf--H_2O-H}$ represent the energies of the system with H on the surface and in the electrolyte, respectively. Details of computing the reaction energy profile have been updated in the revised manuscript.

7. It is suggested to use “ads” or “*” uniformly, in labeling surface adsorbent.

Response: Thanks for your carefully reviewing. The adsorbents have been uniformly labeled by “*”.

8. “Ab initio” should be “ab initio”.

Response: Thank you for your kind suggestion. Following the reviewer's comments, these words have been corrected.

Reviewer #2 (Remarks to the Author):

In this article, Feng et al. reported Pt-induced anti-deactivation of Ru for alkaline hydrogen oxidation reaction. This is a very interesting topic to the fast-moving field of the alkaline HOR as well as anion exchange membrane fuel cells. The results presented are supported by both experimental and theory. The author successfully demonstrated anti-deactivation on Ru/C using 3% Pt by preserving Ru passivation which preserves the interfacial water network for H* oxidation. Therefore I recommend the publication of the work in the prestigious Nature Communications Journal after addressing my comments below.

Response: We are sincerely grateful for the positive comment of the reviewer and the recommendation for publication. Meanwhile, we highly appreciate the reviewer's constructive comments, which have greatly improved our work. We have prepared a point-by-point response to address the raised comments, which are presented below.

1. The authors demonstrated the solid proof using XANES and EXAFS which shows reduced Ru-O which suffices the anti-deactivation provided by Pt. However, Pt is known for the excellent alkaline HOR catalyst well known in the literature which is stable under the HOR (0.0- 1.0 V vs. RHE). For instance, <https://doi.org/10.1039/C4EE00440J>. So, I am not sure the enhancement is coming from Pt itself. It looks like Pt is doing the catalysis at higher HOR potential supported by the synergistic provided by Ru. Clarify this point with an additional control experiment without HOR active components such as TiO₂ or CeO₂.

Response: We sincerely thank the referee for the valuable comment. To exclude the contribution of Pt itself at higher HOR potential, we followed the reviewer's suggestion and synthesized Pt doped TiO₂ and Pt doped CeO₂ catalysts (Fig. R1). For fair comparison, the Pt loadings on these non-active metal oxides are controlled to be the same as the main samples (3%Pt- and 10%Pt-Ru/C) in our manuscript. In other word, the mass loadings of 0.6% and 2%Pt on these oxides are similar to those of 3% and 10%Pt-Ru/C. Based on the XRD patterns in Fig. R1a and R1c, only CeO₂ or TiO₂ related diffraction peaks are observed, while metallic Pt based peaks are absent, excluding the formation of Pt nanoparticles. The HOR performance are further investigated. The 0.6%Pt- and 2%Pt-CeO₂ samples are extremely inert for the catalysis, showing current density even smaller than 0.2 mA cm⁻² within the studied potential region. For 0.6 %Pt-TiO₂, although it possesses HOR activity at high potential, the current density (<1.2 mA cm⁻²) is far less than 3%Pt-Ru/C (~3.2 mA cm⁻²). For 2%Pt-TiO₂, the current density (<2.2 mA cm⁻²) at high potential increases but is still lower than that of 3%Pt-Ru/C and 10%Pt-Ru/C (~3.2 mA cm⁻²). In a word, the Pt decorated non-active metal oxide materials show poor HOR catalytic activity at high potential, demonstrating Pt itself with these mass loadings is not active enough to drive such high current density at high potential. Thus, in this study, the performance enhancement is mainly due to the protected Ru sites instead of Pt itself. We have added a brief discussion in the revised manuscript and supplementary information.

Fig. R1. HOR performance of Pt decorated CeO₂ and TiO₂. XRD patterns of Pt-TiO₂ (a) and Pt-CeO₂ (b). HOR polarization curves of Pt-TiO₂ (c) and Pt-CeO₂ (d). The Pt mass loadings on these metal oxides are controlled to be the same as those on Ru/C. In detail, 0.6%Pt- and 2%Pt- correspond to 3%Pt- and 10%Pt-Ru/C, respectively.

2. What about the stability of the catalyst towards long-term tests to prove the robustness of the catalysts?

Response: We are grateful for the insightful comment. The stability of the catalysts has been examined by chronoamperometry test at +0.15 V vs. RHE (Fig. R2). Within 400 min test, Ru/C loses its initial current density by 56.5%, demonstrating its instability towards HOR test. In comparison, the current density loss is reduced to only 19.3% for 3%Pt-Ru/C under the same condition, verifying the robustness of the 3%Pt-Ru/C catalyst. The result is included in the relevant part of the revised manuscript.

Fig. R2. The stability tests of Ru/C and 3%Pt-Ru/C using RDE method.

3. Why the HOR polarisation is recorded at 2500 rpm and not at 1600 rpm which is typical? Explanation is needed to justify the rpm used in this study.

Response: We appreciate the referee's thoughtful comment. Both 1600 and 2500 rpm are often used rotating speeds for HOR test (J. Am. Chem. Soc. 2022, 144, 25, 11138–11147; Energy Environ. Sci., 2020, 13, 3064-3074; Nat. Commun. 7, 10141 (2016)). In our test, the glassy carbon-based rotating disk electrode contains a PFTE shell, which is ultra hydrophobic and gathers the gas bubbles on the electrode when the rotating speed is not fast enough. As a result, the quality of the signal is not good enough and the test is more difficult. Thus, 2500 rpm is used for the test. For more comprehensive comparison, the LSV curves at 1600 rpm are also compared (Fig. R3), which display the same trend as those obtained at 2500 rpm.

Fig. R3. Polarization curves of the studied materials with the rotating rate of 1600 rpm.

4. In Figure 2a, it will be good to show the full polarization curve from 0.0 V to 0.8 V to reveal the robustness of the anti-deactivation in this study.

Response: We thank the referee for the valuable suggestion. The polarization curves are extended to 0.8 V. Obviously, at the potential as high as 0.8 V, the high current density of 3%Pt-Ru/C is still maintained, further verifying the high robustness of the anti-deactivation strategy (Fig. R4). All the polarization curves with larger region have been updated in the revised manuscript and supplementary information.

Fig. R4. Polarization curves of Ru/C, 3%Pt-Ru/C, Pt/C and Pt/C-com.

5. In Figure 2c, the exchange current density is normalized with the geometric area and not with the electrochemical active surface area (ECSA). In order to see real improvement, the j_0 must be normalized with ECSA.

Response: We appreciate the referee's comment. The ECSA values have been obtained by Cu_{upd} -stripping voltammetry and the results are summarized in Fig. R5. As Pt loading increases, before the Pt aggregates formation (10%Pt-Ru/C), the ECSA remains almost the same, indicating Pt single atom doping onto Ru nanoparticles would not significantly change ECSA. Once Pt aggregates form, the 10%Pt- and 20%Pt-Ru/C show obviously increased ECSA, which might originate from the contribution of small Pt clusters. As Pt content further increases, the ECSA gradually reduces, probably due to the formation of large Pt nanoparticles. To see the real performance improvement, the j_0 is normalized by the ECSA (Fig. R6). A volcano-type trend is observed, where 3%Pt-Ru/C

shows the best ECSA-normalized performance. The discussion is included in the revised manuscript and supplementary information.

Fig. R5. a-h, The Cu_{upd}-stripping tests in Ar saturated 0.5 M H₂SO₄ of the studied materials. i, The calculated ECSA.

Fig. R6. The calculated ECSA-normalized exchange current density of the studied materials.

6. The reason for the improvement of the j_0 of 3% PtRu/C needs to be highlighted in greater detail. Response: Thanks for the referee's constructive comment. Considering j_0 is obtained at the micro-polarization region near 0 V vs. RHE, the chemical state of the samples at this potential is analyzed. According to the valence state information in Fig. 3b, at the applied potential of 0 V vs. RHE, although both Ru/C and 3%Pt-Ru/C mainly exist in metallic state, the oxidation state of Ru in Ru/C is still slightly higher than that in 3%Pt-Ru/C. Meanwhile, based on the coordination information in Fig. 4c, at 0 V vs. RHE, the oxidation-related Ru-O_s is more pronounced in RuC than that in 3%Pt-Ru/C. Thus, compared with 3%Pt-Ru/C, Ru/C is still oxidized to some extent, while the oxidation

would lead to decreased HOR performance, which could account for the higher J_0 of 3%Pt-Ru/C.

7. Figure 2d, Nice plot. Why did the number of electron decrease with 3% PtRu/C with the increase in potential above 0.1 V vs. RHE, unlike Pt/C?

Response: We appreciate the reviewer's thoughtful comment. The decreased number of electron could be attributed to the current contribution of OH* adsorption on Ru surface at ~ 0.1 V vs. RHE (Angew. Chem. Int. Ed. 2017, 56, 15594–15598). To minimize the influence of OH* adsorption, we lowered the scan rate of the polarization curves to 1 mV s^{-1} . At such small scan rate, the contribution of OH* adsorption is limited and the decrease of the electron number is suppressed (Fig. R7). We have updated all the polarization curves obtained using 1 mV s^{-1} scan rate for more accurate comparison.

Fig. R7. Calculated electrons involved in the HOR catalysis based on polarization curves obtained at the scan rate of 1 mV s^{-1} .

8. Have the authors explored above 10% Pt additions as the activity is increased with Pt content (Supplementary Figure S7). Please explore 20 % and 50% to see if we get the improvement further.

Response: According to the reviewer's suggestion, 20%Pt- and 50%Pt-Ru/C were synthesized and tested (Fig. R8). The 20%Pt-Ru/C shows the highest J_0 among the studied materials, while for 50%Pt-Ru/C, lower J_0 is observed. As discussed in the response to the 5th comment, the performance enhancement of 20%Pt-Ru/C is mainly attributed to the increased ECSA, while 3%Pt-Ru/C possesses the highest ECSA normalized HOR activity (Fig. R6). The results of 20%Pt- and 50%Pt-Ru/C have been included in the revised supplementary information.

Fig. R8. a, Polarization curves of Ru/C, 1%Pt-, 3%Pt-, 10%Pt-, 20%Pt- and 50%Pt-Ru/C. b, Exchange current density.

9. It is a good carryout deactivation strategy using Pt. Owing to the sluggish HOR kinetics in alkaline the anode loading is higher in real fuel cell devices. Therefore it is good to include such a strategy on non-noble catalysts such as TiO₂ as reported by Zhou et al. (<https://doi.org/10.1038/s41929-020-0446-9>), where similar activation is achieved using TiO₂. The author should comment on this.

Response: We express our gratitude to the reviewer for the thoughtful comment. We're also interested that confining Ru nanoparticles onto the defective TiO₂ could catalyze HOR with anti-deactivation feature. According to the work reported by Zhou et al., the lattice confinement allows electron transfers from TiO₂ to Ru nanoparticles, thus greatly enhancing the anti-oxidation ability of the material. The less easily oxidized Ru nanoparticles provide ideal surface for HOR catalysis with anti-deactivation. This strategy is innovative and interesting. A concern exists in terms of the low conductivity of metal oxides, which would lead to high ohmic loss in real device. Thus, currently, loading Ru onto carbon support might be more achievable in practical application. We have added a brief discussion in the introduction.

10. What about the real device performance in the anion exchange membrane fuel cell data with stability to solidify the robustness strategy?

Response: We appreciate the referee's valuable comments. We have to admit that we don't have the relevant equipment and experience in testing AEMFC, and therefore the test was conducted in other institutions. Due to the lack of experience and limited time, the testing conditions might not have been well optimized. Fig. R9 displays the H₂/O₂ fuel cell test using the synthesized materials as anode. The fuel cell using 3%Pt-Ru/C as the anode yields a current density of 0.25 A cm⁻² at 0.65 V and a peak power density of 0.25 W cm⁻². In comparison, Ru/C anode shows extremely poor performance and the current density even cannot reach 0.2 A cm⁻². Furthermore, the stability of the fuel cells is tested at fixed current density. For 3%Pt-Ru/C, the AEMFC outputs stable voltage at the current density of 0.2 A cm⁻² for at least 20 h, while for Ru/C, the cell quickly undergoes deactivation within 2.5 h even at smaller current density of 0.15 A cm⁻². Combining the cell activity and stability, it could be concluded that the strategy of doping Pt for anti-deactivation of Ru might be achievable in real device. The AEMFC results have been added to the relevant part of the revised manuscript.

Fig. R9. a, H₂/O₂ AEMFC polarization plots with Ru/C and 3%Pt-Ru/C as the anode catalysts. **b,** Long-term stability test of the fuel cells with 3%Pt-Ru/C anode catalyst at 0.2 A cm⁻² and Ru/C anode catalyst at 0.15 A cm⁻².

11. The operando-XANES and SR-FTIR show solid evidence of the water dynamics at the interface which gives the fundamental understanding of the de-activation of the Ru provided by Pt which appeal to the broader readership enjoyed by Nature Communications.

Response: We here again express our gratitude to the referee for carefully reviewing and the high evaluation of the manuscript.

Reviewer #3 (Remarks to the Author):

Yanyan Fang et al. use combined experimental and computational approach to study the modifications in Ru catalyst that lead to superior catalytic activity toward hydrogen oxidation reaction (HOR) as compared to unmodified catalyst. The reason is attributed to the surface modifications via Pt single atoms that effectively prevent Ru from oxidation. It was further shown that the oxidation of a Ru catalyst disturbs interfacial water network, which is crucial for HOR reaction. While work is interesting with plethora of characterization and computational work that complements the experiments, I have several major concerns about the work.

Response: We sincerely appreciate the reviewer's positive feedback on our work and the constructive comments, which would greatly enhance the quality of the manuscript. Following the reviewer's comments, the point-by-point responses are presented below:

1. While there might be evidence of superior intrinsic activity of Pt-Ru/C samples, this work provides no convincing data that proposed modification of the Ru catalyst would lead to increased durability of the electrochemical system or a device (no time studies).

Response: We appreciate the reviewer's valuable comments. Following the reviewer's suggestion, the durability of the electrochemical system and the anion exchange membrane fuel cell (AEMFC) was performed in the revised manuscript.

First, the stability of the electrochemical system was examined by chronoamperometry test. As shown in Fig. R1, at the applied potential of +0.15 V vs. RHE, Ru/C loses its initial current density by 56.5% within 400 min, demonstrating its instability towards HOR test. In comparison, for 3%Pt-Ru/C, the current density loss is reduced to only 19.3% under the same condition, verifying the robustness of the 3%Pt-Ru/C catalyst.

Fig. R1. The stability tests of Ru/C and 3%Pt-Ru/C using RDE method.

For the durability of the device, an AEMFC was assembled with the synthesized HOR catalysts as anode. Nevertheless, we have to admit that we don't have the relevant equipment and experience in testing AEMFC, so the test was conducted in other institutions. Due to the lack of experience and limited time, the testing conditions might not have been well optimized. Fig. R2 displays the H₂/O₂ fuel cell test using the synthesized materials as anode. The fuel cell using 3%Pt-Ru/C as the anode yields a current density of 0.25 A cm⁻² at 0.65 V and a peak power density of 0.25 W cm⁻². In comparison, Ru/C anode shows extremely poor performance, and the current density even cannot reach 0.2 A cm⁻². Furthermore, the stability of the fuel cells is tested at fixed current density. For 3%Pt-Ru/C, the AEMFC outputs stable voltage at the current density of 0.2 A cm⁻² for at least 20 h, while for Ru/C, the cell quickly undergoes deactivation within 2.5 h even at smaller current density of 0.15 A cm⁻². Combining the cell activity and stability, it could be concluded that the strategy of doping Pt for anti-deactivation of Ru might be achievable in real device.

In a word, the proposed modification of the Ru catalyst would lead to increased durability of the electrochemical system and real device. The data of AEMFC and the RDE stability tests have been updated in the revised manuscript.

Fig. R2. **a**, H₂/O₂ AEMFC polarization plots with Ru/C and 3%Pt-Ru/C as the anode catalysts. **b**, Long-term stability test of the fuel cells with 3%Pt-Ru/C anode catalyst at 0.2 A cm⁻² and Ru/C anode catalyst at 0.15 A cm⁻².

2. There are several elements of the computational work that are unclear and therefore difficult to judge.

a) Why is vacuum introduced in the model (Fig 5a) and how are water molecules in a water layer prevented from drifting into the vacuum during the AIMD simulations? In a fully equilibrium state, one would expect for water molecules to distribute to the whole region above the catalyst surface.
 Response: We sincerely appreciate the reviewer's valuable comments, which could certainly improve the quality of our work. Our work investigates the deactivation mechanism of Ru, which originates from surface oxidation-induced broken interfacial water network based on the operando EXAFS and FTIR results. To better simulate this process, vacuum layer is introduced to the explicit solvent model. On the one hand, the vacuum helps to set periodic boundary condition to simulate an infinite system in a finite box. On the other hand, the vacuum layer allows the water layer to relax to equilibrium state and reduces the boundary effect, which could avoid the unwanted systematic deviation of the box-bound water layer system in the simulation.

In addition, in AIMD simulations, the water layer is generally simulated by ice-like structure, due to the quite similar hydrogen bond network structure in liquid and ice. The unwanted water relaxation to the whole region in AIMD simulations often happens at high temperature. However,

for the ice-like structure at room temperature of 300 K, when the vacuum layer is thick enough to cut off the interaction between water molecules and the lower surface of the slab, the water molecules are stably connected by the hydrogen bond and form regular crystal structure. Thus, water molecules are less likely to distribute to the whole region above the catalyst surface, which is also demonstrated by the stable water layer after 15 ps simulation and the constant temperature and energy during the simulation (Supplementary Fig. 30 and 31). Moreover, the introduction of vacuum to the explicit solvent model has been used in many papers. For example, Li et al. used this kind of model with vacuum to investigate the role of hydrogen bond network connectivity in the kinetic pH effect in hydrogen electrocatalysis on Pt (Nat. Catal. **5**, 900–911 (2022)).

b) How was pH taken into account?

Response: In the original manuscript, the main attention of the theoretical calculation was paid to study the deactivation of Ru and Pt-induced anti-deactivation, and the pH was not considered. To address the reviewer's concern on the pH effect, OH⁻ was further added to the solvent models to simulate the alkaline media and K⁺ was introduced to balance the charge with the pH set to 14 based on previously reported method (J. Phys. Chem. Lett. **13**, 10550–10557 (2022)). The charge density difference maps, the O* and OH* adsorption energy, the concentration distribution profiles of O atoms and the H* oxidative desorption thermodynamics were re-considered based on the system with KOH. Based on the results shown in Fig. R3, the introduced Pt atom could effectively protect Ru surface from oxidative passivation, which preserves interfacial water network and benefits the essential H* oxidative release process. The results show the similar trend with our original manuscript and the similar conclusion could also be obtained, indicating pH is not the key factor influencing the result. However, to more precisely simulate the real reactive environment, we have updated the calculation results with the addition of KOH in the revised manuscript.

Fig. R3. Insights into interfacial behavior. **a**, Representative snapshot of the interfacial structure on Pt-Ru(0001)/H₂O. Grey, red, orange, turquoise balls: H, O, Pt and Ru atoms. **b**, Charge density difference maps of Ru(0001)/H₂O interface and Pt-Ru(0001)/H₂O interface (Isosurface value: 0.002 e Å⁻³; blue: charge consumption; red: charge accumulation). **c**, The adsorption energies of O* and OH* on Ru(0001) and Pt-Ru(0001) surfaces. **d**, Concentration distribution profiles of O atoms along the surface normal direction. The pink shaded area represents adsorbed surface oxidation O, while the blue shaded area stands for the first two interfacial water layers within 4 Å to the corresponding surfaces. **e**, Reaction pathway for a surface-adsorbed H* released to interfacial water layer (Green ball: the H atom involved in the reaction). **f**, Schematic diagram for HOR catalysis on Pt-Ru(0001) surface.

c) When determining thermodynamics of H* desorption, was water structure obtained from equilibrated AIMD trajectories and how? Was this obtained from a snapshot and if so, from which one? Does the choice of exact water structure/water network have a huge effect on the H* desorption thermodynamics?

Response: We thank the referee for the insightful comments. When determining the H* desorption thermodynamics, the water structure was obtained from the equilibrated AIMD trajectories after 15 ps simulation. Then, the structure was further optimized by DFT calculation to achieve a stable water structure for the subsequent energetic calculation.

To understand whether the choice of exact water structure/water network has a huge effect on the H* desorption thermodynamics, we further obtained the different structures at 9 ps and 12 ps during the AIMD simulations and these structures were further optimized by DFT for the energetic study. The optimized structures are presented in Supplementary Fig. 36-38, which show different water network structures. Based on the thermodynamic results shown in Fig. R4, the different water structures from different snapshots will lead to different H* releasing thermodynamic energy barrier. However, the total trend remains the same that the introduction of Pt would facilitate H* desorption while the oxidized Ru(0001)-O_{1/3ML} surface always possesses hindered H* desorption. This result further demonstrates the conclusion in our manuscript that the difference in H* releasing energy barrier is highly associated with the interfacial water molecule number and denser surface water content could benefit H* release from the surface to the electrolyte and thereby improve the HOR performance.

Fig. R4. The thermodynamic energy barriers for H* release from surface to electrolyte based on the water structure obtained at 9ps, 12 ps and 15ps.

d) When determining barriers on Fig 5e, were and how were transition states confirmed to be transition states?

Response: We sincerely appreciate the reviewer's valuable comments. Fig. 5e displays the thermodynamic process where an adsorbed H* on the metal surface is desorbed and released to the interfacial water network. It can be regarded as a proton coupled electron transfer (PCET) process ($H^* \rightarrow H^+ + e^-$), which is a thermodynamic process mainly driven by limiting potential (Chem. Rev. 2019, 119, 7610–7672). Therefore, we focused on the thermodynamic energy barrier instead of the complex kinetic process. Based on the calculation results, the thermodynamic energy barrier is helpful enough to unveil the influence of interfacial water on the HOR activity and could correspond well with the experimental results. Thus, the transition state is not considered in our study.

e) How are plots on Figure 5d obtained to determine the number of water molecules above the metal surfaces – from a snapshot or is this derived as an average from the production run?

Response: We are grateful for the reviewer's thoughtful comments. For the O concentration profiles, the 10000 structures within the 5-15ps AIMD simulation (timestep: 1fs) were counted and the average result was shown in Fig. 5d.

f) Structures in Fig 5c and Fig 5e are very difficult to understand and it is therefore very difficult to

understand the exact role of water in the HOR mechanism. Although Figure 5c and 5e look nice, it would be useful to show the same structures from these Figures but in larger format in SI.

Response: Thank you for your carefully reading the manuscript. Following the reviewer's suggestion, the same structures as Fig. 5c and 5e have been shown in larger format in the revised supplementary information.

Reviewer #4 (Remarks to the Author):

Response: We appreciate the precious comments raised by the referee. We hope the reviewer can get satisfied with our responses.

REVIEWER COMMENTS

Reviewer #1 (Remarks to the Author):

The authors have responded appropriately to the comments on the original version of manuscript, as well as made necessary revisions in the resubmitted work. The quality of the the resubmitted manuscript is high and acceptable for publication.

As a minor concern for consideration, the authors are suggested to highlight the OH- group (with different colors) in Fig. 5a to clearly illustrate its position in the slab.

Reviewer #2 (Remarks to the Author):

The authors address my comment in full. A few minor points need to be clarified before publication.

1. I am not sure why the author is not able to achieve good performance with Pt-CeO, Pt-TiO₂, or even Ru/TiO₂ systems. Few works exist in the literature showing comparable or better performance for alkaline HOR. For instance,

<https://doi.org/10.1021/acs.chemmater.0c02048>, <https://doi.org/10.1002/smll.202307497>,

<https://doi.org/10.1038/s41929-020-0446-9>. Justify your answer.

2. The author has done the chronoamperometry test to show the robustness of the catalyst (Figure R2). Still 19.3 % deactivation can be seen in 400 min which is quite significant. What is the cause of this degradation?

3. In Figure R3, I can see the nice data with 1600 rpm. Higher RPM creates turbulence.

4. In Figure R8, I see the comparison based on geometric area. The jo normalized with ECSA offers fair comparisons among various catalysts. Figures R6 and R8 should be included in the main manuscript which gives a broader spectrum of the Pt ratio.

5. The AEMFC performance is below the state-of-the-art performance shown in the literature (<https://link.springer.com/article/10.1007/s10008-022-05261-4>). Further optimization may improve the fuel cell data significantly. However, the additional AEMFC data demonstrate the robustness of the concept which is commendable.

Reviewer #3 (Remarks to the Author):

Authors have performed a number of additional experiments to address all reviewer comments and concerns. They have addressed all my comments and questions in a satisfactory manner, bringing more value and clarity to the paper.

Reviewer #4 (Remarks to the Author):

Reviewer #1 (Remarks to the Author):

The authors have responded appropriately to the comments on the original version of manuscript, as well as made necessary revisions in the resubmitted work. The quality of the the resubmitted manuscript is high and acceptable for publication.

As a minor concern for consideration, the authors are suggested to highlight the OH⁻ group (with different colors) in Fig. 5a to clearly illustrate its position in the slab.

Response: We again thank the referee for the valuable and insightful comments that have certainly improved the quality of our manuscript. We are also glad that the referee is satisfied with our responses and recommends the manuscript for publication. Following the reviewer's suggestion, the OH⁻ group in the interfacial water is highlighted with different color for better illustration in the revised manuscript.

Reviewer #2 (Remarks to the Author):

The authors address my comment in full. A few minor points need to be clarified before publication.

Response: We sincerely appreciate the referee's valuable comments and the precious time spent to review the manuscript. According to the referee's comments, the point-by-point responses are presented below.

1. I am not sure why the author is not able to achieve good performance with Pt-CeO₂, Pt-TiO₂, or even Ru/TiO₂ systems. Few works exist in the literature showing comparable or better performance for alkaline HOR. For instance, <https://doi.org/10.1021/acs.chemmater.0c02048>, <https://doi.org/10.1002/sml.202307497>, <https://doi.org/10.1038/s41929-020-0446-9>. Justify your answer.

Response: We appreciate the referee's valuable comment and we are glad to clarify this issue. In response to the reviewer's original comment that "I am not sure the enhancement is coming from Pt itself. It looks like Pt is doing the catalysis at higher HOR potential supported by the synergistic provided by Ru", Pt decorated TiO₂ and CeO₂ were synthesized. For fair comparison, the Pt mass loadings on TiO₂ and CeO₂ were controlled to be the same as the main samples in our manuscript. As summarized in Table R1, 3%Pt-Ru/C and 10%Pt-Ru/C contain 0.6 wt% and 2 wt% Pt, respectively. Thus, 0.6 wt% and 2 wt% Pt was introduced to CeO₂ and TiO₂. The synthesized Pt-TiO₂ and Pt-CeO₂ possess ultra-low Pt loading and such low Pt decoration cannot provide effective conducting network, which leads to the low HOR activity. In the mentioned papers, the mass loadings of Ru, Pd and Pt are much higher (Table R1) in the range of 10.9 wt% to 93 wt%, which could be the reason for the high HOR activities. Thus, it is unfair to directly compare the HOR performance of Pt-TiO₂ and Pt-CeO₂ in our work with the materials possessing much higher platinum-group metal mass ratio.

Table R1. Summarized mass ratios of the platinum-group metal in the catalysts.

Sample	Mass ratio	Reference
3%Pt-Ru/C	0.6 wt% Pt	This work
10%Pt-Ru/C	2 wt% Pt	
0.6%Pt-TiO ₂	0.6 wt% Pt	This work

2%Pt-TiO ₂	2 wt% Pt	
0.6%Pt-CeO ₂	0.6 wt% Pt	
2%Pt-CeO ₂	2 wt% Pt	
Ru@TiO ₂	10.9 wt% Ru	https://doi.org/10.1038/s41929-020-0446-9
PtRu/TiO ₂ /C-MW	21.3 wt% Pt, 11.3 wt% Ru	https://doi.org/10.1002/sml.202307497
Pt-Ce ₄	93 wt%	https://doi.org/10.1021/acs.chemmater.0c02048
Pt-Ce ₂₀	78 wt%	
Pd-Ce ₄	57 wt%	
Pd-Ce ₂₀	40 wt%	

2. The author has done the chronoamperometry test to show the robustness of the catalyst (Figure R2). Still 19.3 % deactivation can be seen in 400 min which is quite significant. What is the cause of this degradation?

Response: We are grateful for the reviewer's insightful comment. Generally, the stability test on RDE is not good enough, which is a common phenomenon also observed in other papers even for the commercial Pt/C (*Nat. Commun.* **13**, 5894 (2022); *J. Mater. Chem. A*, **8**, 10168(2020); *J. Am. Chem. Soc.* **145**, 22069–22078 (2023)). This unsatisfying stability on RDE could be attributed to the catalyst detachment from the electrode, the irreversible metal leaching, the metal aggregation and the carbon corrosion under rotating (*Nat. Catal.* **5**, 363–373 (2022)). However, Ru/C shows much larger current decrease (56.5%) than 3%Pt-Ru/C (19.3%). Considering both Ru/C and 3%Pt-Ru/C have been synthesized under the same condition and possess the same metal loadings, morphologies and particle sizes, the RDE results could indicate that the oxidation-induced intrinsic catalyst deactivation of Ru/C is much more severe than 3%Pt-Ru/C. Besides, the AEMFC test could further demonstrate the higher stability of 3%Pt-Ru/C than Ru/C.

3. In Figure R3, I can see the nice data with 1600 rpm. Higher RPM creates turbulence.

Response: We sincerely appreciate the referee's comment. Following the referee's comment, the polarization curves obtained at 1600 rpm are compared and displayed in the revised supporting information as Supplementary Figure 8. The trend is similar with the data collected at 2500 rpm, which further confirms the main scope of the manuscript that the introduction of Pt could protect Ru from oxidative deactivation. Meanwhile, both 1600 rpm and 2500 rpm are significantly lower than the upper limit of the rotating speed of RDE to avoid the turbulent flow (Allen J. Bard, Larry R. Faulkner. *Electrochemical methods: fundamentals and applications*, chapter 9.3).

4. In Figure R8, I see the comparison based on geometric area. The j_o normalized with ECSA offers fair comparisons among various catalysts. Figures R6 and R8 should be included in the main manuscript which gives a broader spectrum of the Pt ratio.

Response: Thanks for the reviewer's comment. Figure R8 displays the geometric area-normalized exchange current densities and Figure R6 displays the ECSA-normalized exchange current densities. Following the reviewer's suggestion, the data in Figure R6 and Figure R8 are included in the main manuscript as Fig. 2b and 2c.

5. The AEMFC performance is below the state-of-the-art performance shown in the literature

(<https://link.springer.com/article/10.1007/s10008-022-05261-4>). Further optimization may improve the fuel cell data significantly. However, the additional AEMFC data demonstrate the robustness of the concept which is commendable.

Response: We thank the reviewer for the insightful comment. We agree that the AEMFC performance in our work is below the state-of-the-art. As we have discussed before, the test was not fully optimized because we don't have the relevant equipment and experience in testing AEMFC and the tests were conducted in other institutions. Additionally, there are a lot of other important technological parameters, such as selection of carbon substrate, ink preparation, catalysts loading, cell assembling and testing conditions, which could affect the device performance. Moreover, the key point of this work is to reveal the anti-deactivation mechanism of Pt-Ru/C system for HOR catalysis, and the device performance is not the main target. In our future related work, we will follow the reviewer's valuable suggestion and spend more efforts to optimize the device performance by collaborating with fuel cell groups.

Reviewer #3 (Remarks to the Author):

Authors have performed a number of additional experiments to address all reviewer comments and concerns. They have addressed all my comments and questions in a satisfactory manner, bringing more value and clarity to the paper.

Response: We are glad that the reviewer is satisfied with our responses. We thank the reviewer again for the thoughtful and valuable comments, which effectively improves the quality of our manuscript.

Reviewer #4 (Remarks to the Author):

Response: We appreciate the reviewer's valuable comments and the time spent to review our manuscript.